# Autofocus Retrieval: An Effective Pipeline for Multi-Hop Question Answering With Semi-Structured Knowledge

**Derian Boer**                                                                          *boer@uni-mainz.de*
*Institute of Computer Science*
*Johannes Gutenberg University Mainz*

**Stephen Roth**

**Stefan Kramer**                                                          *kramer.informatik@uni-mainz.de*
*Institute of Computer Science*
*Johannes Gutenberg University Mainz*

**Reviewed on OpenReview:** *https://openreview.net/forum?id=U2vqruHfQY*

## Abstract

In many real-world settings, machine learning models and interactive systems have access to both structured knowledge, e.g., knowledge graphs or tables, and unstructured content, e.g., natural language documents. Yet, most rely on either. Semi-Structured Knowledge Bases (SKBs) bridge this gap by linking unstructured content to nodes within structured data. In this work, we present Autofocus-Retriever (AF-Retriever), a modular framework for SKB-based, multi-hop question answering. It combines structural and textual retrieval through novel integration steps and optimizations, achieving the best zero- and one-shot results across all three STaRK QA benchmarks, which span diverse domains and evaluation metrics. AF-Retriever's average first-hit rate surpasses the second-best method by 32.1%. Its performance is driven by (1) leveraging exchangeable large language models (LLMs) to extract entity attributes and relational constraints for both parsing and reranking the top-$k$ answers, (2) vector similarity search for ranking both extracted entities and final answers, (3) a novel incremental scope expansion procedure that prepares for the reranking on a configurable amount of suitable candidates that fulfill the given constraints the most, and (4) a hybrid retrieval strategy that reduces error susceptibility. In summary, while constantly adjusting the focus like an optical autofocus, AF-Retriever delivers a configurable amount of answer candidates in four constraint-driven retrieval steps, which are then supplemented and ranked through four additional processing steps. An ablation study and a detailed error analysis, including a comparison of three different LLM reranking strategies, provide component-level insights that are valuable for advancing the model and for enabling researchers and users to adapt, optimize, or extend its parts. The source code is publicly available at `https://github.com/kramerlab/AF-Retriever`.

## 1 Introduction

Large Language Models (LLMs) have rapidly evolved in recent years, demonstrating impressive performance on a range of complex tasks. As a result, they are increasingly integrated into everyday workflows, scientific research, and decision-making processes. However, despite their capabilities, LLMs remain prone to significant limitations. They can produce inaccurate or biased outputs, fail to recognize gaps in their own knowledge, and generate biased or fabricated ("hallucinated") content that is difficult to detect (Bender et al., 2021; Feldman et al., 2023). Their effectiveness in reasoning-based tasks is also the subject of ongoing scrutiny (Kambhampati, 2024). To mitigate these shortcomings, Retrieval-Augmented Generation (RAG) systems have been introduced. These systems enhance LLMs by retrieving relevant external information

before generating responses, improving factual accuracy and robustness. Beyond RAG, many contemporary AI systems, including those for question answering (QA), typically leverage either unstructured content like natural language documents or structured data, such as knowledge graphs and relational tables. The latter provides precise, queryable information, while the former offers broad, context-rich knowledge that is retrieved via dense embedding-based search. However, in many real-world domains, both types of information are available and valuable.

This has led to the development of Semi-Structured Knowledge Bases (SKBs), which combine the advantages of both by linking unstructured documents to nodes in knowledge graphs. SKBs thus enable new methods of knowledge access and inference by bridging the gap between purely symbolic and contextual information. Recent work by Wu et al. (2024b) exemplifies the potential of SKBs. They constructed SKBs across three real-world domains and assembled the STaRK benchmarks of corresponding multi-hop QA datasets. These datasets reflect realistic scenarios where answering a question requires both relational reasoning over structured elements and contextual interpretation of unstructured texts. In response, several specialized and hybrid approaches have been proposed to leverage SKBs for complex QA tasks. However, many existing methods focus on specific techniques in isolation, overlooking opportunities to integrate complementary ideas.

To address this gap, we introduce Autofocus-Retriever (AF-Retriever), a zero-shot, multi-strategy method for QA over SKBs. The central idea is to iteratively adjust the scope and focus of candidate entities and relations by combining structured querying with neural retrieval and multiple ranking stages, including one final re-ranking. AF-Retriever synthesizes established components based on this idea into a cohesive pipeline that dynamically balances precision and recall. In a zero- and one-shot setting, it outperforms state-of-the-art methods (SOA) on all STaRK datasets. Only one recent preprint reports superior results on a single benchmark. However, that approach relies on domain-specific fine-tuning, whereas our method operates without task-specific training.

The proposed solution includes the following steps, which are described in section 3 in more detail. (i) *Target type prediction:* The first simple, yet effective step identifies which entity type is suitable as an answer and constitutes the first layer of focus. (ii) *Formalization as Cypher queries:* It further takes advantage of the inherent ability of common LLMs to interpret natural language questions and translate them into Cypher (Francis et al., 2018) queries, a popular language for querying property graphs. These queries include an updated target node label and several relational triplets with constants, variables, and their attributes to encode multi-hop questions. Our extended analysis confirms that even smaller, open base models are able to produce Cypher queries off the cuff. (iii) *Querying:* Our analysis shows that including and omitting relational facts in node embeddings can increase or decrease their entity retrieval utility depending on its application. To retrieve semantically relevant entities as constants to be considered, we employ Vector Similarity Search (VSS) without relational data embedded. (iv) *Entity retrieval:* Building on extracted relational triplets and symbol candidates, a variant of graph-based path search prefilters answer candidates by their relational information. This search, implemented as a sequence of set intersections, narrows down answer candidate sets to only those that meet extracted conditions. (v) *Scope expansion:* Because several nodes are eligible for a constant (e.g., "University of Miami", "Miami University", or "Miami Dade College" for the search string "Miami Uni"), AF-Retriever incrementally increases the number of candidates for constants again so that the relational constraints remain fulfilled and the number of false positives stays small, but still a predefined number of answers is returned. This novel approach dynamically balances the specificity and sensitivity of entity retrieval, a challenge in itself. (vi) *Merging and sorting:* By merging candidate sets from both the aforementioned neuro-symbolic and a more naive vector-based retrieval, AF-Retriever mitigates potential errors introduced by individual retrieval steps. Our parameterized analysis demonstrates that, while the purely vector-based retrieval performs much worse than SOA methods, the infusion of a few of its answers into the candidate pool can notably increase the overall performance. (vii) *Reranking:* Finally, with a focus on detailed unstructured information, AF-Retriever leverages LLMs to rerank the top-$k$ candidates by either scoring or mutually comparing their relevance and coherence with the question. To identify the verifiable most suitable reranking strategy for SKB-based question answering, this work contributes a theoretical and experimental comparison of pointwise, pairwise, and listwise reranking with LLMs.

The remainder of this paper is organized as follows: Section 2 surveys prior work related to question answering over Semi-Structured Knowledge Bases. Section 3 outlines the proposed methodology in detail. Section 4 reports on our experimental evaluation including a comparison with SOA systems and studies of ablation, hyperparameter sensitivity, error propagation, and latency. Finally, Section 5 concludes the paper and discusses directions for future research.

## 2    Related work

Recent surveys have outlined the synergy between Large Language Models (LLMs) and knowledge graphs (KGs), emphasizing both the opportunities and challenges that arise when they are integrated (Agrawal et al., 2024; Pan et al., 2024). Pan & et al. (2023) further highlight the ongoing difficulties in integrating structured reasoning with the generative capabilities of LLMs, pointing to key research directions for improving factual accuracy, context retention, and reasoning over hybrid knowledge sources.

**Retrieval across textual and relational knowledge**  Our work falls within the domain of retrieval-based, multi-hop question answering (QA) over semi-structured knowledge, where both structured (relational) and unstructured (textual) data must be effectively combined. Wu et al. (2024b) introduced the concept of a Semi-Structured Knowledge Base (SKB), linking knowledge graph (KG) nodes with associated free-text documents. Their STaRK benchmark suite evaluates methods designed to operate over such hybrid corpora and includes experiments with the following six baseline techniques. Among these retrieval methods, one of the simplest is *Vector Similarity Search (VSS)*, which embeds a query and candidate entities, each represented by concatenated textual and relational information, into a shared space for cosine similarity computation. *Multi-Vector Similarity Search* extends this by separating and embedding text and structure separately, enabling a finer-grained comparison. *Dense Passage Retriever* (Karpukhin et al., 2020) trains query and passage encoders using QA pairs to optimize retrieval accuracy. *QA-GNN* (Yasunaga et al., 2021) tackles the problem by constructing a joint graph consisting of QA elements and KG nodes. It performs relevance scoring using LLM-generated embeddings, followed by message passing over the graph to facilitate joint reasoning. *VSS+Reranker* (Chia et al., 2024; Zhuang et al., 2024) uses a language model to assign a relevance score to each of the top-$k$ VSS results, refining the initial retrieval with semantic ranking grounded in an SKB context. This corresponds to steps 7 and 8 with a pointwise reranker and $\alpha = 0$ in our pipeline (see below). *AVATAR* (Wu et al., 2024a) represents an agent-based approach, in which a comparator module generates contrastive prompts for optimizing LLM behavior during tool usage. Its performance study includes *ReAct* (Yao et al., 2023), which blends reasoning with next action decision in an interleaved LLM prompt, and Reflexion (Shinn et al., 2023), which employs episodic memory to refine agent decisions across tasks. *HybGRAG*, a complementary line of work presented by Lee et al. (2025), combines multiple retrievers with a critic module to refine answer generation across both structured and textual data. *4StepFocus* (Boer et al., 2024) augments VSS+Reranker by two preprocessing steps: Triplet generation for extraction of relational data by an LLM and substitution of constants and variables in those triplets to narrow down answer candidates. AF-Retriever shares this aspect, but it additionally follows a hybrid retrieval approach, features different LLM reranking strategies, and its greedy-like, VSS-based candidate search is more advanced than the string similarity search of 4StepFocus. Xia et al. (2025) propose *KAR*, a query expansion method that integrates relational constraints from the KG into the retrieval pipeline and performs listwise LLM reranking concatenating $n$ responses with filtered triplets as the final expansion appended to the original query.

**Retrieval with integrated domain- and task-specific fine-tuning of employed LLMs**  Further methods fine-tune LLMs on training sets from the same distribution as test sets before conducting the retrieval procedure. At the cost of LLM interchangeability and applicability when suitable training sets are missing, this increases the potential of higher accuracy. MoR (Mixture of Structural-and-Textual R) (Lei et al., 2025) and mFAR (multi-Field Adaptive Retrieval) (Li et al., 2024) address semi-structured retrieval by decomposing documents into multiple fields, each indexed independently. *MoR* frames retrieval in terms of three stages: planning, reasoning, and organizing, bridging graph traversal and text matching through path-based scoring. Thereby, data are decomposed into fields and each field is scored separately against the query using lexical- and vector-based scorers. The reranking component adaptively predicts the importance of a field by conditioning on the document query. *mFAR* (Li et al., 2024) also adaptively weights fields

based on the query and generates the entire planning graph in one shot, avoiding repeated LLM calls. In contrast, *PASemiQA* (Yang et al., 2025) repeatedly consults an agent which decides which action to take next, then performs the action to retrieve relevant information from the SKB that is used to determine the next action. In contrast to our incremental scope-expanding approach, if no node for an entity is found via keyword matching, a fixed number of VSS-based candidates is taken in each case.

Property graph databases such as Neo4j and JanusGraph have gained widespread adoption in both academia and industry over the past decade. Cypher, a prominent query language for such systems, allows expressive querying of node and relationship patterns using constructs like MATCH, WHERE, and RETURN (Francis et al., 2018). Most recently, *GraphRAFT* (Clemedtson & Shi, 2025) was released in a preprint including evaluation on two STaRK test sets. It combines different fine-tuned LLMs with a Cypher engine. The first LLM identifies entities in an user query which are inserted in three different templates for one- and two-hop queries with up to two triplets. Another fine-tuned LLM selects the most suited queries, before they are executed to generate a joint subgraph. A third LLM ranks entities within this subgraph. The authors claim remarkable results on the synthetically generated PRIME dataset but also acknowledge the dependency on data-specific fine-tuning, while our approach leverages the ability of base LLMs to generate Cypher queries from natural language without fine-tuning. Additionally, without adaptation, AF-Retriever is more flexible as it does not build on a fixed query template with a restricted number of triplets and supports more syntax.

Appendix A.4 lists these works including the base LLMs they employed, Section 4 compares our results of AF-Retriever against them on the STaRK benchmark suite.

**Additional perspectives**  Beyond retrieval-focused QA systems, several recent efforts have tackled related challenges such as mitigating hallucinations or enhancing interpretability. Guan et al. (2024) proposed a knowledge graph-based retrofitting method to reduce LLM hallucinations using automatically generated triplets. Wang et al. (2024) use embedding-based techniques to enable QA across multiple documents. Vedula & Parthasarathy (2021) present FACE-KEG, a KG-transformer model for fact verification with an emphasis on explainability. Kundu & Nguyen (2024), as well as Shahi (2024), explore approaches for identifying and structuring false claims within a KG framework for fact checking. Similarly, Shakeel & Jain (2021) take a pipeline approach combining KG embeddings and traditional NLP for factual verification. Knowledge Solver (Feng et al., 2023) builds whole subgraphs first, allowing the LLM to select entities in subsequent path-finding steps, starting from a randomly selected term in the question. Mountantonakis & Tzitzikas (2023) provide a semi-automatic web application that assists users in comparing facts generated by ChatGPT with those in KGs by using multiple RDF KGs for enriching ChatGPT responses. While it has a few parallels to our work, the web interface requires manual interpretation.

**Contributions of this work**  In addition to the integration and optimization of several neuro-symbolic core concepts (VSS, text2cypher parsing, triplet-based graph search, and LLM reranking), this work introduces multiple novel approaches to SKB-based question answering, such as: an incremental procedure to dynamically balance specificity and sensitivity, the introduction of a two-model ensemble strategy, and different LLM reranking approaches to SKB-based question answering. Our modular pipeline, AF-Retriever, outperforms SOA methods. An extensive ablation study and error analysis provide valuable insights for future work.

## 3  Autofocus-Retriever

### 3.1  Problem definition and notations

A Semi-Structured Knowledge Base $SKB = (\mathbb{V}, \mathbb{E}, \mathbb{D})$  (Wu et al., 2024b) consists of a knowledge graph $\mathcal{G} = (\mathbb{V}, \mathbb{E})$ and associated text documents $\mathbb{D} = \bigcup_{v \in \mathbb{V}} D_v$, where $\mathbb{V}$ is a set of graph nodes and $\mathbb{E} \subseteq \mathbb{V} \times \mathbb{V}$ a set of their connecting edges, which may be directed or undirected, and weighted or unweighted. Given an SKB, a query $q$ in natural language, and a set of candidates for query answers $\mathbb{C} \subseteq \mathbb{V}$, our question answering scenario expects a list of (up to 20) most fitting candidates to $q$ in descending order.

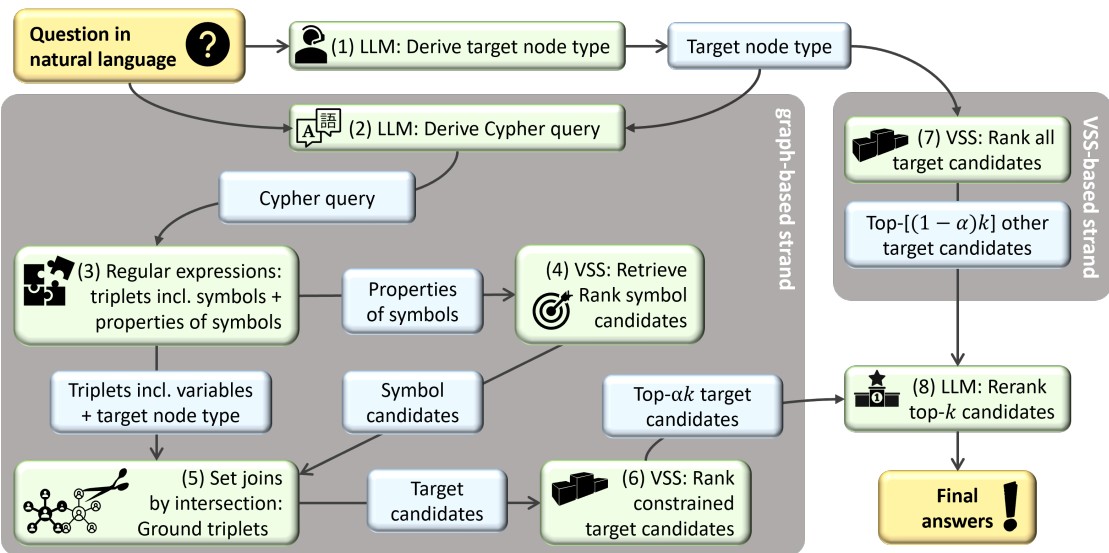

Figure 1: Overview of Autofocus-Retriever's processing from the user query $q$ to the final answers. The green boxes represent its eight steps, the blue boxes represent the objects passed between them.

---

**Algorithm 1** AF-Retriever

**Input:**
a Semi-Structured Knowledge Base $SKB = (\mathbb{V}, \mathbb{E}, \mathbb{D})$, a query $q$ in natural language, candidates for query answers $\mathbb{C} \subseteq \mathbb{V}$, number of candidates to rank and return $k$, an upper bound for the number of candidates per constant to consider $l_{max}$, the graph-based retrieval weight $\alpha \in [0, 1]$

**Output:** A list of $k$ answers to $q$ in descending order

1: $y.\text{type} \leftarrow \text{DERIVE\_TARGET\_TYPE}(q, \mathbb{V}.\text{types})$          ▷ step 1
2: $q'_{\text{cypher}} \leftarrow \text{DERIVE\_CYPHER}(q, \mathbb{V}.\text{types}, \mathbb{E}.\text{types}, y.\text{type})$     ▷ step 2
3: $y_{\text{cypher}}.\text{type}, \mathbb{T}, \mathbb{S}_{raw} \leftarrow \text{REGEX}(q'_{\text{cypher}}, \mathbb{V}.\text{types}, \mathbb{E}.\text{types})$    ▷ step 3
4: $\mathbb{S} \leftarrow \text{RETRIEVE\_SYMBOL\_CANDIDATES}(\mathbb{S}_{raw}, \mathbb{V}, \mathbb{D}, l_{max})$    ▷ step 4
5: $\mathbb{C}_{\text{cypher}} \leftarrow \emptyset$                                    ▷ start of step 5
6: $l \leftarrow 1$
7: **while** $|\mathbb{C}_{\text{cypher}}| < k$ & $l < l_{max}$ **do**
8:     $\mathbb{C}_{\text{cypher}} \leftarrow \text{GROUND\_TRIPLETS}(\mathbb{T}, S[: l], y_{\text{cypher}}.\text{type}, \mathbb{E})$   ▷ Algorithm 2
9:     $l \leftarrow l^{1.5} + 0.5$
10: **end while**                                 ▷ end of step 5
11: $\boldsymbol{y}_{\text{cypher}} \leftarrow \text{VSS}(q, \mathbb{C}_{\text{cypher}}, \alpha k, SKB)$            ▷ step 6
12: $\boldsymbol{y}_{\text{vss}} \leftarrow \text{VSS}(q, \mathbb{C}(y.\text{type}) \backslash \mathbb{C}_{\text{cypher}}, k - |\boldsymbol{y}_{\text{cypher}}|, SKB)$   ▷ step 7
13: $\boldsymbol{y} \leftarrow \boldsymbol{y}_{\text{cypher}} + \boldsymbol{y}_{\text{vss}}$                     ▷ concatenate arrays
14: $\boldsymbol{y} \leftarrow \text{RERANK}(\boldsymbol{y}, SKB)$                     ▷ step 8
15: **return** $\boldsymbol{y}$

---

Before we explain the individual steps of our approach in more detail, we introduce the following notions: Let $\mathbb{V}.\text{types} = \bigcup_{v \in \mathbb{V}} type(v)$ be the set of all existing node types in SKB, where $type(v)$ represents the node type (e.g., "protein") of any node $v \in \mathbb{V}$. Let $y.\text{type} \in \mathbb{V}.\text{types}$ be a predicted node type of the target variable $y$, and $\mathbb{C}(y.type) \subseteq \mathbb{C}$ the set of all candidate nodes $y$ with $type(y) = y.\text{type}$. $NEIGHBORS(v, e.\text{type}, \mathbb{E}, +)$ returns the union of all nodes $u$ for which an edge of type $e.\text{type} \in \mathbb{E}.\text{types}$ exists that points from $u$ to $v$. $NEIGHBORS(v, e.\text{type}, \mathbb{E}, -)$ denotes the union of all nodes $u$ for which an edge of type $e.\text{type} \in \mathbb{E}.\text{types}$ exists that points in the other direction – from $v$ to $u$. A distinction is often made between variables and constants. While constants typically represent a single, fixed value or specific entity, variables act as placeholders for one or more unknown values or entities. In this work, we refer to constants as a small, finite set of candidate nodes that we incrementally extend for entities with a name or title attribute that can be

used for a database scan. We define variables as entities without a searchable name or title. We use the term symbols for sets of both constants and variables. This approach acknowledges the potential for ambiguity in certain contexts. For instance, the constant "Mexico" could semantically refer to either the country or the city (i.e., Mexico City). Let $S$ be a dictionary, assigning a sorted array of candidate nodes $\boldsymbol{s}(x)$ to each symbol $x$. $S[:l]$ denotes a clone of $S$, where each assigned array $\boldsymbol{s}(x)$ is replaced by a set of its first $l$ nodes. $k$ and $l_{max}$ are hyperparameters, which put a ceiling on the entity search. $\alpha \in [0,1]$ weighs between two retrieval strategies. These hyperparameters are described below in more detail.

## 3.2 Pipeline of AF-Retriever

Figure 1 provides a high-level overview of the key components of our framework and the flow of intermediates at its interfaces. Algorithm 1 formalizes the eight steps of AF-Retriever. Each of these steps calls one of the functions explained below with a running example.

**Step 1) Target type prediction:** As the first step, line 1 of Algorithm 1, DERIVE_TARGET_TYPE($q$, $\mathbb{V}$.types) is called, which lets an LLM predict the node type of any answer to the question $q$. In essence, the LLM is prompted: "An instance of which of the given entity types $\mathbb{V}$.types could correctly answer the query $q$?" The full templates of all LLM prompts used in our implementation are provided in Appendix A.2. This filter step constitutes the first layer of focus and is considered in both strands of the hybrid framework.

---

Example input (above the line) and output (below the line) of DERIVE_TARGET_TYPE:

$q=$ "Which research in molecular biology has been produced by a Miami uni in 2015?"
$\mathbb{V}$.types $=$ ["paper","author","institution","field of study"]

---

$y$.type $=$ "paper"

---

**Step 2) Cypher query extraction:** Subsequently, in line 2, DERIVE_CYPHER($q$, $\mathbb{V}$.types, $\mathbb{E}$.types, $y$.type) uses an LLM to extract queryable information from $q$. In summary, the LLM is prompted: "Return a Cypher query derived from the given query $q$, ensuring that the following restrictions are met: [...]" As central part of these restrictions, the prompt includes the available node and edge types $\mathbb{V}$.types and $\mathbb{E}$.types (called "labels" in Cypher) as well as $y$.type from the previous step. This inclusion is important to separate the relational information that can be queried in an SKB from non-relational entity features included in $q$ that might be only available in the text documents $\mathbb{D}$ but not in the knowledge graph $\mathcal{G}$. Our extended analysis in section 4 confirms that even smaller, open base models are able to produce Cypher queries making instructions for a custom machine-readable format redundant.

---

Output of DERIVE_CYPHER($q$, $\mathbb{V}$.types, $\mathbb{E}$.types, $y$.type) for the running example:

$q'_{\text{cypher}} =$
"*MATCH (i:institution {name: 'Miami uni'})<-[:employed_at]-(a:author)-[:wrote]->(p:paper)
p-[:has_field_of_study]->(f:field_of_study {name: 'molecular biology'})
WHERE p.publication_year = 2015
RETURN p.title*"

---

**Step 3) Parsing with regular expressions:** Instead of executing $q'_{\text{cypher}}$ in a Cypher-driven database directly, AF-Retriever parses $q'_{\text{cypher}}$ into a structured format via nested regular expressions in line 3. This detour enables more controlled downstream processing and supports exchangeable LLMs that usually cannot create vector search-based Cypher queries. $\text{REGEX}(q'_{\text{cypher}}, \mathbb{V}.\text{types}, \mathbb{E}.\text{types})$ extracts the target variable's node type $y_{\text{cypher}}.\text{type} \in \mathbb{V}.\text{types}$, a set of relational triplets $\mathbb{T}$, and node attributes $\mathbb{S}_{raw}$ from $q'_{\text{cypher}}$, based on Cypher keywords like "MATCH", "WHERE", "RETURN", "CONTAINS", and other standardized syntax. Since the LLM may violate the restriction of returning a variable of type $y.\text{type}$, $y_{\text{cypher}}.\text{type}$ does not necessarily equal $y.\text{type}$ from step 1. Each triplet $\tau \in \mathbb{T}$ is a tuple $(h, e, t)$ of an edge type $e \in \mathbb{E}.\text{types}$ and two symbols. A symbol is either categorized as a constant if a name or title attribute is given, and as a variable otherwise. Each attribute in $S_{raw}$ is a tuple of a symbol ID, its node type, an attribute name $p \in \mathbb{V}.\text{attr}$, and an attribute value. REGEX can filter out invalid or unsupported components from $q'_{\text{cypher}}$.

---

Output of $\text{REGEX}(q'_{\text{cypher}}, \mathbb{V}.\text{types}, \mathbb{E}.\text{types})$ for the running example:

$y_{\text{cypher}}.\text{type} = $ "paper"
$\mathbb{T} = \{(a, \text{"employed at"}, i), (a, \text{"wrote"}, p), (p, \text{"has field of study"}, f)\}$
$S_{raw} = \{(a, \text{"type"}, \text{"author"}), (p, \text{"type"}, \text{"paper"}), (f, \text{"type"}, \text{"field\_of\_study"}), (i, \text{"type"}, \text{"institution"}), (p, \text{"publication year"}, =2015), (f, \text{"name"}, \text{"molecular biology"}),$
$(i, \text{"name"}, \text{"Miami uni"})\}$

---

**Step 4) Symbol candidates retrieval:** $\text{RETRIEVE\_SYMBOL\_CANDIDATES}(\mathbb{S}_{raw}, \mathbb{V}, \mathbb{D}, l_{max})$ in line 4 then retrieves $l_{max}$ candidates for each *constant* and filters an unlimited number of candidates for *variables* with at least one attribute. The "title" and "name" attributes of each constant are used as their individual search strings, which will be used later. Every attribute that is neither "title" nor "name" is applied as a hard filter for candidates for each symbol. For example, "year $\geq 2012$" rules out all papers published before 2012. If a filter cannot be applied because an attribute name or value is unavailable in the SKB and it belongs to a constant, then this attribute is appended to its search string. Finally, RETRIEVE\_SYMBOL\_CANDIDATES sorts each constant's candidates by their vector cosine similarity to the embedding of the constant's search string, then cuts the top $l_{max}$ out. To illustrate why more than one candidate for a constant may be necessary, consider the simple example of searching for a paper by "J. Smith" published in the journal "Nature". Ambiguous references like this can yield multiple plausible, yet contextually inaccurate candidates for both constants. For instance, the first hit for "J. Smith" might have no "Nature" publication, while the second or third hit does. Regarding the entity and relation types, we implement two contrary variants to customize restrictiveness: a) Strict mode: Apply a hard filter on nodes and edges by their predicted entity and relation types, b) lenient mode: Accept any annotated node or edge type with a high vector similarity for higher error tolerance against incorrect entity and relation type classifications.

---

Output of RETRIEVE\_SYMBOL\_CANDIDATES for the symbol i of the running example, $l_{max} = 3$:

$\boldsymbol{s}(i) = [\text{"University of Miami"}, \text{"Miami University"}, \text{"Miami Dade College"}]$

---

**5) Triplets grounding:** The purpose of lines 5–10, given the output of the previous steps, is to identify at least $k$ answers (candidates for the target variable $y$), while keeping the total number of candidates for the target variable and other symbols low to maintain precision. The objective is to focus on $k$ answers, because this hyperparameter corresponds to the predefined number of candidates that are handled by a reranker in the last step. To achieve this balanced scope, the triplets are first grounded with the first hit for each constant ($l = 1$). Then, the focus is expanded by gradually incrementing the number of candidates $l$ per constant until either $k$ answers are found or the limit $l_{max}$ is reached. To reduce the number of steps and thus computation, $l$ is incremented exponentially. In the strict mode, candidates are also filtered by edge types. See Algorithm 2 in Appendix A.3 for details on GROUND_TRIPLETS($\mathbb{T}, S[: l], y_{\text{cypher}}.\text{type}, \mathbb{E}$), which returns the candidate set for $y$ depending on $l$ in each iteration. In summary, each triplet $(h, e, t)$ refines the candidate sets for each symbol by propagating constraints bidirectionally via set intersections. This loop continues until convergence (no further eliminations), and the final target candidate set $\mathbb{C}_{\text{cypher}}$ is returned.

---

Output of GROUND_TRIPLETS($\mathbb{T}, S[: l], y_{\text{cypher}}.\text{type}, \mathbb{E}$) for the running example:

**Output for $l = 1$:**
$\mathbb{C}_{\text{cypher}} = \emptyset$

**Output for $l = 2$:**
$\mathbb{C}_{\text{cypher}} =$[{type:"paper", title: "RNA Transcription", year:2015, ...},
{type:"paper", title: "Review on Ribosomes", year:2015, ...}, ...]

---

**Steps 6 and 7) Vector similarity search:** Steps 6 and 7 both rank nodes based on their vector cosine similarity to the search string. In both cases, our VSS routine accepts a search string, a candidate set, and a parameter that specifies the number of candidates to return.

Step 6 applies VSS to the question $q$, the candidates $\mathbb{C}_{cypher}$, and round($\alpha k$) as the number of candidates to return in line 11. To increase the chance of hits in a two-model ensemble fashion, step 7 retrieves round($(1 - \alpha)k$) additional top candidates retrieved independently of the Cypher-derived constraints, by calling VSS for the entire candidate set $\mathbb{C}$ in line 12. If $y$.type is a valid node type, step 7 only considers nodes of this type.

---

Example output:

**Step 6) VSS**
$y_{\text{cypher}} =$[{type:"paper", title: "Review on Ribosomes", year:2015, ...},
{type:"paper", title: "RNA Transcription", year:2015, ...}, ...]

**Step 7) VSS**
$y_{\text{vss}} = \mathbb{C}_{\text{cypher}} =$[{type:"paper", title: "Biodiversity in Miami", year:2015, ...},
{type:"paper", title: "RNA Transcription", year:2015, ...}, ...]

---

Table 1: Alternative LLM reranking approaches implemented in AF-Retriever for $k$ candidates.

| | main characteristic | expected #prompts | expected #tokens |
|---|---|---|---|
| pointwise reranker | one prompt per item assigning a confidence score between 0 and 1 to each item | $k$ | $O(k)$ |
| listwise reranker | one prompt including all entities requesting a sorted list of them | $1$ | $O(k)$ |
| pairwise reranker | binary insertion sort using one prompt for each pairwise comparison during sorting | $k * \log(k)$ | $O(k * \log(k))$ |

**Step 8) LLM Reranking:** Finally, after combining $\boldsymbol{y}_{cypher}$ and $\boldsymbol{y}_{vss}$ into a unified vector $\boldsymbol{y}$ (line 13), the top-$k$ answers are reranked (line 14). Although reranking does not alter the set of top-$k$ answers, it adjusts their order to increase the number of first hits based on the complete information available for the top candidates. We implement three LLM-based reranking options: pointwise, pairwise, and listwise. The *listwise* reranker sends a single prompt containing all top-$k$ candidates, along with their node attributes, annotated documents, 1-hop relations, and 2-hop relations where the relation type to the candidate is 1-1 or n-1. The LLM is then instructed to order the candidates according to how well they answer $q$. This straightforward approach requires only one prompt with $O(k)$ tokens, but it demands an LLM with a sufficient large context window and memory to handle all $k$ descriptions simultaneously. The *pairwise* reranker repeatedly prompts the descriptions of only two candidates to an LLM at once and asks which candidate answers the question better. Assuming transitivity, $O(k * \log k)$ tokens need to be sent, but a smaller LLM context window is required because less information has to be processed at once. The *pointwise* reranker, as employed by Wu et al. (2024b), presumes that LLMs can produce comparable confidence scores for individual candidates. Our extended experiments show that while the pointwise alternative improves ranking quality over VSS, it performs worse than the two differential (pairwise and listwise) variants on all datasets.

Our implementation of all rerankers provides for three scenarios: 1) In the default case, the node description in every prompt includes all 1-hop and the selected 2-hop relations described above. 2) If the context window is exceeded, the candidate's edges (relations) that are not incident to any retrieved non-target entity candidate are removed from the prompt. 3) If the prompt still exceeds the context limit, all relations are omitted entirely. In practice, even for the listwise reranker, the context window is only rarely exceeded (see Appendix A.6.7). See Table 1 for a theoretical comparison of the reranking approaches, Appendix A.6.7 for an experimental comparison, and Appendix A.2 for the applied prompts.

---

Output of RERANK($\boldsymbol{y}$, SKB) for the running example:

$\boldsymbol{y} =$[{type:"paper", title: "RNA Transcription", year:2015, ..., source: $\boldsymbol{y}_{cypher}$},
{type:"paper", title: "Review on Ribosomes", year:2015, ..., source: $\boldsymbol{y}_{cypher} + \boldsymbol{y}_{vss}$}, ...]

---

# 4 Evaluation

## 4.1 Overall Question Answering Performance

**Datasets** We evaluate AF-Retriever on the three STaRK QA benchmark datasets (Wu et al., 2024b): PRIME, MAG, and AMAZON, each paired with a semi-structured knowledge base (SKB). Each STaRK benchmark provides a set of synthetic training, validation, and test queries, as well as a smaller set of human-generated questions. These benchmarks simulate realistic scenarios by combining relational and textual knowledge with diverse, natural-sounding question formats. The SKBs vary substantially: *PRIME (medical domain)* has the richest relational structure, with diverse node and edge types, and a strong focus on relations. *AMAZON (product recommendation)* includes fewer node types but a large proportion of unstructured textual data (i.e., reviews). *MAG (academic paper search)* strikes a balance, featuring multi-hop relational queries as well as textual attributes like paper abstracts. Each SKB contains hundreds of thousands to millions of nodes and edges. In PRIME, every node is a potential answer. In MAG and AMAZON, only all nodes of specific types (papers and products, respectively) qualify. For additional details on the SKBs and datasets, we refer readers to Appendix A.1 and Wu et al. (2024b).

**Experiment settings** We report standard retrieval metrics averaged over the test queries: *hit@m*: The mean of a binary indicator of whether at least one relevant item is among the top-$m$ results. *recall@m*: The mean proportion of relevant items among the top-$m$. *mrr*: The **m**ean **r**eciprocal of the **r**ank at which the first relevant answer appears, with $m$ being the maximum of every rank. AF-Retriever is evaluated in a zero-shot scenario with fixed default hyperparameters and pretrained LLMs. However, Subsection 4.2 includes a hyperparameter sensitivity analysis, and Section 5 points out training potential for optimization and fine-tuning.

We use the GPT OSS 120B, one of the largest and popular open-weight models that still fits on a single H100 GPU, as the LLM backbone, and OpenAI's text-embedding-3-small model for the vector-based semantic search of AF-Retriever. We set the maximum number of candidates per symbol to $l_{max} = 100$, which allows for a broad scope but keeps the runtime within an acceptable range. In compliance with the LLM reranker of Wu et al. (2024b), $k = 20$ answers are accepted for reranking. The hyperparameter $\alpha$ is 2/3

Table 2: Performance of zero- and one-shot benchmark methods including AF-Retriever averaged over the three synthetic and human-generated STaRK benchmark test sets. See Section 2 and Appendix A.4 for full method names, references, and used base LLMs. A dash indicates that no result is reported for one or more datasets for the concerned metric. The best results are marked in bold, the second-best are underlined. "*" denotes results that were only published for the 10%-version of the synthetically generated test sets.

| STaRK test sets | synthetically generated | | | human-generated | | |
|---|---|---|---|---|---|---|
| metric (in percent) | hit@1 | hit@5 | mrr | hit@1 | hit@5 | mrr |
| BM25 | 27.8 | 46.9 | 36.7 | - | - | - |
| VSS (Ada-002) | 27.0 | 47.9 | 36.8 | 28.5 | 46.8 | 38.3 |
| VSS (Multi-Ada) | 27.0 | 49.7 | 37.3 | - | - | - |
| DPR | 10.1 | 35.0 | 21.3 | 7.6 | 19.4 | 14.1 |
| VSS + Reranker* | 34.7 | 55.5 | 43.7 | 39.9 | 55.5 | 46.4 |
| QA-GNN | 16.1 | 36.8 | 27.2 | - | - | - |
| ToG | - | - | - | - | - | - |
| AvaTaR | 37.6 | 55.2 | 45.5 | 41.6 | 56.9 | 48.5 |
| 4StepFocus* | 46.9 | 63.3 | 54.6 | 52.9 | 61.5 | 50.3 |
| KAR | 45.0 | 62.5 | 53.0 | 52.6 | 63.9 | 57.6 |
| HybGRAG | - | - | - | - | - | - |
| ReAct | 29.5 | 48.7 | 38.1 | 31.6 | 48.4 | 40.3 |
| Reflexion | 32.6 | 51.5 | 41.6 | 31.5 | 45.5 | 39.8 |
| AF-Retriever | **62.0** | **76.7** | **68.4** | **55.8** | **66.4** | **60.7** |

Table 3: Performance of zero- and one-shot benchmark methods including AF-Retriever on the synthetically generated test sets. "*" denotes results that were only published for the 10%-version of the synthetically generated test sets.

| STaRK test set | PRIME | | | MAG | | | AMAZON | | |
|---|---|---|---|---|---|---|---|---|---|
| metric (in percent) | hit@1 | hit@5 | mrr | hit@1 | hit@5 | mrr | hit@1 | hit@5 | mrr |
| BM25 | 12.8 | 27.9 | 19.8 | 25.8 | 45.2 | 34.9 | 44.9 | 67.4 | 55.3 |
| VSS (Ada-002) | 12.6 | 31.5 | 21.4 | 29.1 | 49.6 | 38.6 | 39.2 | 62.7 | 50.4 |
| VSS (Multi-Ada) | 15.1 | 33.6 | 23.5 | 25.9 | 50.4 | 36.9 | 40.1 | 65.0 | 51.6 |
| DPR | 4.5 | 21.8 | 12.4 | 10.5 | 35.2 | 21.3 | 15.3 | 47.9 | 30.2 |
| VSS+[pointw.]Reranker* | 18.3 | 37.3 | 26.6 | 40.9 | 58.2 | 49.0 | 44.8 | 71.2 | 55.7 |
| QA-GNN | 8.8 | 21.4 | 14.7 | 12.9 | 39.0 | 29.1 | 26.6 | 50.0 | 37.8 |
| ToG | 6.1 | 15.7 | 10.2 | 13.2 | 16.2 | 14.2 | - | - | - |
| AvaTaR | 18.4 | 36.7 | 26.7 | 44.4 | 59.7 | 51.2 | 49.9 | 69.2 | 58.7 |
| 4StepFocus* | 39.3 | 53.2 | 45.8 | 53.8 | 69.2 | 61.4 | 47.6 | 67.6 | 56.5 |
| KAR | 30.4 | 49.3 | 39.2 | 50.5 | 65.4 | 57.5 | 54.2 | 68.7 | 61.3 |
| HybGRAG | 28.6 | 41.4 | 34.5 | 65.4 | 75.3 | 69.8 | - | - | - |
| ReAct | 15.3 | 32.0 | 22.8 | 31.1 | 49.5 | 39.2 | 42.1 | 64.6 | 52.3 |
| Reflexion | 14.3 | 35.0 | 24.8 | 40.7 | 54.4 | 47.1 | 42.8 | 65.0 | 52.9 |
| AF-Retriever (ours) | **46.2** | **63.7** | **54.0** | **78.6** | **91.4** | **84.0** | **61.2** | **75.2** | **67.3** |

Table 4: Performance on the synthetically generated test sets with training data used.

| STaRK test set | PRIME | | | MAG | | | AMAZON | | |
|---|---|---|---|---|---|---|---|---|---|
| metric (in percent) | hit@1 | hit@5 | mrr | hit@1 | hit@5 | mrr | hit@1 | hit@5 | mrr |
| PASemiQA | 29.7 | - | 31.0 | 43.2 | - | 50.2 | 45.9 | - | 55.7 |
| mFAR | 40.9 | 62.8 | 51.2 | 49.0 | 69.6 | 58.2 | 41.2 | 70.0 | 54.2 |
| MoR | 36.4 | 60.0 | 46.9 | 58.2 | 78.3 | 67.1 | **52.2** | **74.6** | **62.2** |
| GraphRAFT | **63.7** | **75.4** | **69.0** | **71.1** | **85.3** | **76.9** | - | - | - |

rounded, which favors the graph-based retrieval over the mainly vector-based strand of Algorithm 1, but still incorporates a reasonable number of the most vector-similar answers of the predicted node type. An analysis in Subsection 4.2 suggests that the results presented here can still be further improved slightly if $\alpha$ was optimized on a validation set, with maxima between 0.4 and 0.85. For step 8, we select the pairwise reranker with binary insertion sort. It requires $\log(k)$ more tokens than the listwise approach in total but information is be processed in smaller chunks.

**Zero-shot results** Table 2, compares the average performance of AF-Retriever with the reported results of other baseline and SOA methods (including both peer-reviewed publications and preprints) on the three synthetically- and human-generated test sets. The results on the synthetic test sets are broken down further for each dataset in Table 3. Both tables are constrained to zero- and one-shot methods, which do not require domain-specific training data. Due to the small size of the human-generated test sets (81-98 compared to 1642-2801 QA pairs) and the limited availability of published results of other methods, we present our detailed results for them in Appendix A.5 only. AF-Retriever achieves state-of-the-art performance across all three synthetic STaRK benchmarks, outperforming all other zero- and one-shot baselines in terms of hit@1, hit@5, and mrr clearly. For instance, with regard to hit@1, AF-Retriever outperforms the second-best method (KAR) on the AMAZON dataset by 12.9%. On PRIME, it surpasses the runner-up (4StepFocus) by 17.6%, and on MAG it beats the second-best method (HybGRAG) by 20.2%.

**Comparison against supervised methods and fine-tuned systems** Table 4 summarizes the performance of supervised systems that require dataset-specific training and typically rely on fine-tuning. Despite this training advantage, AF-Retriever retains its superiority in most settings. The notable exception is the

Table 5: Ablation study of AF-Retriever. Target node type prediction is omitted for AMAZON and MAG because all questions target the same type. While "step 1 + 7" omits the graph-based retrieval, "steps 1 to 7" describes the hybrid variant with $\alpha = 2/3$. For "steps 1 to 6", in between, the metrics are calculated based on the chronological retrieval order because the answers are actually not yet ordered in another way.

| dataset | PRIME | | | | MAG | | | | AMAZON | | | |
|---|---|---|---|---|---|---|---|---|---|---|---|---|
| metric in % | h@1 | h@20 | r@20 | mrr | h@1 | h@20 | r@20 | mrr | h@1 | h@20 | r@20 | mrr |
| steps 1 + 7 | 15.1 | 48.9 | 34.1 | 24.0 | 32.2 | 68.3 | 44.4 | 42.3 | 42.1 | **81.3** | **35.7** | 53.0 |
| steps 1 to 5 | 23.7 | 46.8 | 33.4 | 29.6 | 24.0 | 72.0 | 44.7 | 37.1 | 5.7 | 19.1 | 9.8 | 8.8 |
| steps 1 to 6 | 30.5 | 52.4 | 37.6 | 36.5 | 51.7 | 80.6 | 51.5 | 61.8 | 15.0 | 28.1 | 14.6 | 18.6 |
| steps 1 to 7 | 33.8 | **70.9** | **51.2** | 42.5 | 59.5 | **94.3** | **61.4** | 71.3 | 35.6 | 80.5 | 35.5 | 45.1 |
| steps 1 to 8 | **46.2** | **70.9** | **51.2** | **54.0** | **78.6** | **94.3** | **61.4** | **84.0** | **61.2** | 80.5 | 35.5 | **67.3** |

PRIME QA set, where GraphRAFT achieves a 39.7% higher hit@1 rate than AF-Retriever. This result indicates that interpretation of highly specialized medical entities, characterized by numerous and diverse node and edge types, is particularly challenging for out-of-the-box LLMs, making supervised learning particularly beneficial. Our extended analysis across multiple LLMs in the subsequent subsection supports this interpretation. On PRIME, retrieval quality is more strongly affected by the choice of underlying language model, whereas on the other datasets performance differences between models are comparatively small. Moreover, GraphRAFT's hit@1 rate drops from 44.2% to 14.7% on the PRIME validation set when fine-tuning is omitted (Clemedtson & Shi, 2025).

If one was wiling to spend effort on the optimization of selected critical hyperparameters of AF-Retrieval, like $\alpha$, moderate performance gains could still be achieved (hit@20 rate by 1.0, 0.4, and 0.1 percentage points on the three datasets, see Fig. 2 and the next section). However, systematic optimization and fine-tuning across the eight pipeline components are beyond the scope of this work, as our study is restricted to a zero-shot setting.

## 4.2 Further Analyses

**Ablation study** Our ablation results are summarized in Table 5. The first row covers the purely VSS-based retrieval with target type prediction if applicable. The next two rows represent the graph-based, neuro-symbolic retrieval, without and with VSS-based ranking of target candidates. The steps 1 to 5 are summarized here, because they are interlocked and only deliver target candidates in combination. Details of the individual steps' performance and error propagation are addressed in the next two paragraphs. The last two lines show the combination of both retrieval strands without and with LLM-based reranking, representing the complete hybrid method.

This study confirms the contribution of each step, because recall@20 steadily increases from step 5 to 7 until the LLM reranker further improves hit@1 and hit@5. Contrary to the trend, step 7 has more hits than steps 2 to 6 on AMAZON regarding hit@20 in 0.2% of the test pairs, and even better than their combination with the non-optimized setting of $\alpha = 2/3$. The following error propagation analysis reveals that while the recall of step 5 is 68% and 97% on PRIME and MAG, respectively, if at least one target candidate is returned, it is only 46% for AMAZON, probably due to the focus on unstructured knowledge. Besides this exception, the hybrid ensemble (steps 1 *to* 7) performs considerably better than either retrieval strategy (step 1 + 7 vs. steps 1 to 6). Finally, the reranker substantially increases the number of first hits and the reciprocal ranks in each case, but hit@1 is still much lower than hit@20, its upper bound.

**Hyperparameter sensitivity** This paragraph highlights the effect of different design and hyperparameter choices for each step from the target type prediction to reranking, summarizing the major findings. Comprehensive results with detailed tables are provided in Appendix A.6.

*Target node type prediction (steps 1 and 2, Appendix A.6.1):* Since all answer candidates of MAG and AMAZON are of the same node type, we only validate the target type prediction on PRIME. Of eight

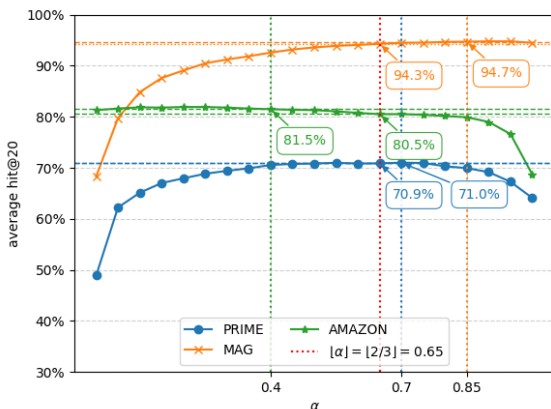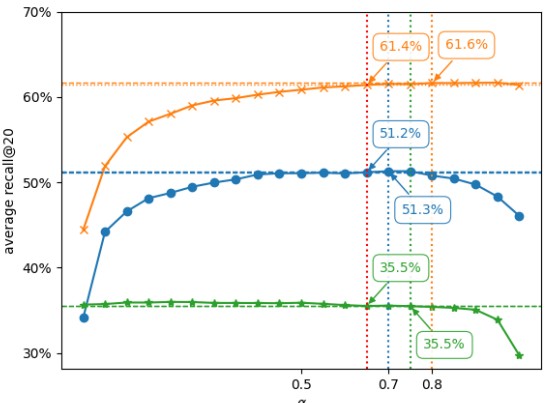

Figure 2: hit@20 and recall@20 of AF-Retriever on the full test sets depending on $\alpha$. The dotted line in red and annotated points represent $\alpha = \lfloor 2/3 \rfloor$ used in the zero-shot evaluation, the other dotted lines mark the optimal $\alpha$ on 10% of the validation sets.

LLMs, only one produced any invalid answers in our experiment, i.e., more than one node type. In 5 out of 280 cases (1.3%), the prediction ($y$.type) of GPT OSS 120B and Gemma 3 27B, the best performing LLMs on this task is incorrect, confusing disease with effect/phenotype for instance. These errors increase after step 2 and 3, where the target type $y_{\text{cypher}}$.type is parsed from the predicted Cypher query. Only GPT-5 mini and GPT OSS 120B produce syntactically correct answers for each test question and predict 97.8% to 98.2% of the target types correctly.

*Strict vs. lenient parsing mode (step 3):* Lenient parsing, which does not filter symbol candidates by their predicted node type, yields the worst results across all datasets. On MAG, hit@20 decreases from 95.6% to 86.1%, while the delta is smaller than 2% on the other datasets. More surprisingly, considering only node types, but ignoring edge types, leads to better performance on PRIME than considering both, although the SKB contains contradictory edges, e.g., "upregulates" and "downregulates". It appears that VSS compensates for this laxity in practice, where the types are included in embedded node descriptions.

*Inclusion vs. exclusion of relations in text embeddings of nodes (steps 4, 6, and 7, Appendix A.6.4):* We observe that including relational information (up to two hops) in node descriptions before embedding them makes a difference, especially for step 7, which uses no preselection of candidates. Inclusion is crucial for direct retrieval tasks because questions with relational parts cannot be answered without it. However, as our experiments confirm, this additional information can be misleading when used to search for non-target entity candidates (step 3), which reduces performance on PRIME and MAG.

*Omitting ambiguous node types (step 2, Appendix A.6.5):* Missing links or entities pose a major challenge when working with knowledge graphs. For instance, in the AMAZON SKB, products are linked to categories in a one-to-one relationship. However, considering other possible taxonomies, one product could be assigned to many categories in different subjective ways. While removing "field of study" for MAG from the list of node types in the LLM prompt of step 2 lifts hit@20 from 88% to 96%, hiding the node type "category" for AMAZON does not confirm our assumption and slightly reduces the performance.

*Stopping criterion for graph-based retrieval (step 5, Appendix A.6.3):* While on PRIME, the overall pipeline reaches its peak for hit@20 at $l_{max} = 100$, the optimal value for $l_{max}$ is 10 on MAG and 1 on AMAZON.

*Balancing graph-based and vector-based retrieval strand (step 7):* Figure 2 plots the hit@20 and recall@20 rate of AF-Retriever on the test sets depending on $\alpha$. Although performance can vary noticeably at the edge cases, typically favoring the graph-based strand of the framework, $\alpha$ between 0.4 and 0.85 demonstrates robustness to the precise value across all datasets. Still, by applying the optimal values on the validation sets

Table 6: First hits in percent of the pointwise, listwise, and pairwise with different LLMs, ordered by their number of weight parameters.

|  | PRIME | | | MAG | | | AMAZON | | |
|---|---|---|---|---|---|---|---|---|---|
|  | pointw | listw | pairw | pointw | listw | pairw | pointw | listw | pairw |
| GPT-5 mini | 39.3 | 47.3 | 48.2 | 53.8 | 82.3 | 77.1 | 43.5 | 54.5 | **60.4** |
| GPT OSS 120B | 41.1 | 47.8 | **49.1** | 55.3 | **83.1** | 71.8 | 48.1 | 47.4 | **60.4** |
| LLaMa 4 Scout Instr. | 26.8 | 24.6 | 34.4 | 51.9 | 60.2 | 65.4 | 40.3 | 35.7 | 53.2 |
| Gemma 3 27B | 31.2 | 22.8 | 42.9 | 52.6 | 54.5 | 70.3 | 46.1 | 42.9 | 55.2 |
| Mistral Small 3.2 Instr. | 26.8 | 29.0 | 39.3 | 51.5 | 62.0 | 80.1 | 44.2 | 42.2 | 50.6 |
| GPT OSS 20B | 32.6 | 41.5 | 42.0 | 59.0 | 76.7 | 76.3 | 42.9 | 46.8 | 57.1 |
| Qwen3 14B | 38.4 | 38.8 | 46.0 | 52.6 | 78.2 | **83.1** | 45.5 | 48.7 | 57.8 |
| Phi-4 | 25.0 | 29.9 | 31.7 | 50.4 | 62.4 | 68.8 | 39.6 | 29.2 | 43.5 |

Table 7: Occurrences of context window overflows, average number of prompts, and average sum of sent and received tokens for different rerankers. Incident (a) of a test QA pair is one if at least one prompt with node descriptions including complete 1-hop relations, and 2-hop relations where the relation type to the candidate is 1-1 or n-1, exceeds the reranker LLM's context window. Incident (b) of a test pair is one if at least one prompt with node descriptions including only those relations to retrieved non-target candidates also exceeds the limit. $k = 20$.

| dataset | reranker | incident (a) | incident (b) | avg. number of prompts | avg. input tokens | avg. output tokens |
|---|---|---|---|---|---|---|
| PRIME | pointwise | 0.0 | 0.0 | $k$ | 52,244 | 5,741 |
|  | listwise | 5.4 | 0.9 | 1 | 35,265 | 2,256 |
|  | pairwise | 0.0 | 0.0 | $3.0k$ | 276,357 | 37,122 |
| MAG | pointwise | 0.0 | 0.0 | $k$ | 23,900 | 5,608 |
|  | listwise | 1.1 | 0.0 | 1 | 16,563 | 2,005 |
|  | pairwise | 0.0 | 0.0 | $3.0k$ | 125,402 | 34,076 |
| AMAZON | pointwise | 0.0 | 0.0 | $k$ | 25,211 | 4,752 |
|  | listwise | 0.0 | 0.0 | 1 | 20,803 | 2,743 |
|  | pairwise | 0.0 | 0.0 | $3.1k$ | 137,379 | 24,856 |

($\alpha$ =0.4, 0.7, and 0.85 for AMAZON, MAG, and PRIME respectively, see Figure 5 in Appendix A.6.6), the hit@20 rate would increase by 1.0, 0.4, and 0.1 percentage points, respectively. Across all datasets, recall@20 varies by less than 0.2 percentage points in the optimal range $\alpha \in [0.65, 0.8]$.

*Comparison of reranking strategies (step 8):* In most cases, all three LLM-based reranking strategies (pointwise, pairwise, and listwise) improve the ranking on the validation sets. Thereby, their first hit rate and their costs differ largely, also depending on the underlying LLM. Table 6 contrast the first hit rate of the reranking approaches for each LLM that we tested for the target type and Cypher prediction already. Each LLM proceeds with its own intermediate results for an end-to-end evaluation, although different models can be used for each step. See Appendix A.6.1 for their properties and performance on previous step and Appendix A.6.7 for hit@5 and mrr.

Independently from the employed LLMs, the pairwise approach consistently delivers the best results in terms of first hits. There is one exception: On MAG, the listwise approach outperforms it with GPT-5 mini, GPT OSS 120B and GPT OSS 20B. However, the pairwise reranker with Qwen 3 14B performs equally well on MAG with a much smaller model size. Generally, GPT OSS 120B, the model with the largest known number of parameters in our experiments, achieves top results across all datasets. Its maximum input length is sufficient even for listwise reranking in most cases, as Table 7 shows. Whether pointwise or listwise reranking works better depends on the dataset and LLM. Table 7 additionally reveals that the

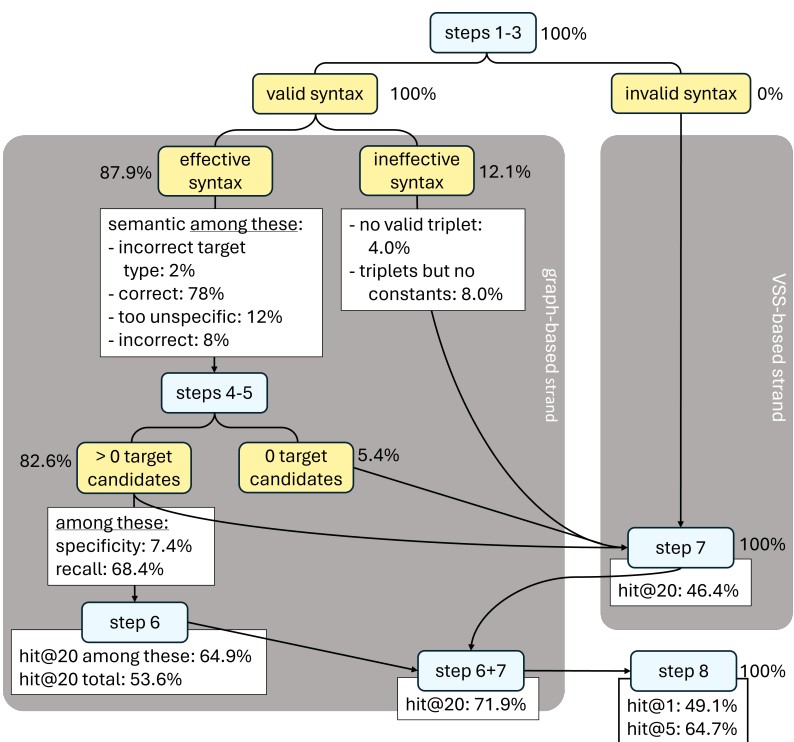

Figure 3: Error propagation for the PRIME validation set. All percentages refer to the total amount, unless otherwise stated.

listwise approach requires about 60 ($3k$) times less prompts and six to eight times less input tokens than the pairwise approach. This is especially noteworthy, because AF-Retriever's running time is heavily dominated by the LLM latency and reranker choice, as addressed in the paragraph after next.

**Error propagation**   Figure 3 illustrates the propagation of errors along the pipeline using the PRIME validation set as a representative example. Corresponding results for MAG and AMAZON, including additional examples, are provided in Appendix A.7.

In the Cypher query generation step, GPT OSS 120B flawlessly produces syntactically valid queries (see Table 14 in Appendix A.6.1). However, 4% of the queries contain no syntactically valid triplet, and in 8% no triplet includes a constant, rendering graph search unsuitable. These queries are not necessarily incorrect, because several questions in MAG and many in AMAZON are non-relational, and hence, such that their answers cannot be constrained through knowledge graph filtering. But not only queries of the form RETURN y.title, as an extreme example, are syntactically valid but semantically underspecified. Others are more specific, but add wrong conditions. While such queries do not violate structural constraints, they fail to uniquely characterize the intended target. As semantic correctness cannot be reliably evaluated automatically, we conduct a manual statistical quality assessment of 50 generated queries containing at least one triplet and one constant per dataset. They are categorized as (i) correct, (ii) not unspecific, and (iii) semantically incorrect. Of these, 78% are judged semantically correct, 12% overly unspecific, and 8% semantically incorrect (e.g., due to incorrect labels or properties). Measurably, in 5.4% of 87.9% with at least one constant and triplet, the generated queries yield no target candidates, terminating the graph-based strand without results. For the remaining cases, steps 4 and 5 achieve a specificity of 7.4% at a recall of 68.4% for $l_{max} = 100$. Among the 87.9% of queries containing at least one triplet and constant, 5.4% return no target candidates, thereby terminating the graph-based branch without results. For the remaining cases, steps 4 and 5 achieve a specificity of 7.4% at a recall of 68.4% for $l_{max} = 100$. After ranking the retrieved candidates using VSS, 53.6% of all questions (64.9% of those answered) contain at least one relevant entity among the top 20 that is merged with the VSS-based branch, which alone achieves a hit@20 of 46.4%.

Table 8: Average, median, and maximum runtimes of AF-Retriever's pipeline steps on the validation sets.

| Step | latency-relevant variant | PRIME | | | MAG | | | AMAZON | | |
|------|--------------------------|-------|------|------|-----|------|------|--------|------|------|
| | | mdn | mean | max. | mdn | mean | max. | mdn | mean | max. |
| 1 | - | 0.40 | 0.67 | 2.98 | 0.00 | 0.00 | 0.00 | 0.00 | 0.00 | 0.00 |
| 2 | - | 0.74 | 0.94 | 5.67 | 0.64 | 0.94 | 5.46 | 0.57 | 0.82 | 7.18 |
| 3 | - | 0.00 | 0.00 | 0.00 | 0.00 | 0.00 | 0.00 | 0.00 | 0.00 | 0.00 |
| 4 | - | 0.10 | 0.29 | 5.10 | 2.44 | 3.26 | 17.36 | 0.16 | 4.51 | 27.38 |
| 5 | $l_{max} = 3$ | 0.01 | 0.05 | 1.96 | 0.13 | 0.23 | 7.08 | 0.01 | 0.12 | 2.52 |
| 5 | $l_{max} = 100$ | 0.07 | 0.24 | 5.96 | 0.10 | 0.23 | 7.75 | 0.01 | 0.12 | 2.86 |
| 6 | - | 0.06 | 0.08 | 0.47 | 4.61 | 4.01 | 5.10 | 0.00 | 3.88 | 11.52 |
| 7 | - | 0.04 | 0.06 | 0.43 | 2.62 | 2.70 | 3.13 | 2.21 | 4.13 | 10.52 |
| 8 | pointwise | 9.28 | 11.28 | 62.55 | 7.57 | 9.06 | 63.61 | 7.21 | 8.45 | 63.87 |
| 8 | pointwise (parall.) | 0.72 | 3.11 | 63.47 | 0.64 | 1.66 | 59.93 | 0.89 | 1.75 | 60.66 |
| 8 | listwise | 2.69 | 4.31 | 87.44 | 1.63 | 2.76 | 50.38 | 1.24 | 2.24 | 25.73 |
| 8 | pairwise | 43.63 | 49.62 | 177.33 | 28.24 | 31.83 | 126.59 | 24.53 | 27.84 | 84.45 |
| sum fastest pipeline | | 2.21 | 5.68 | - | 11.28 | 13.26 | - | 3.86 | 15.45 | - |
| sum slowest pipeline | | 45.04 | 51.9 | - | 38.65 | 42.97 | - | 27.48 | 41.3 | - |

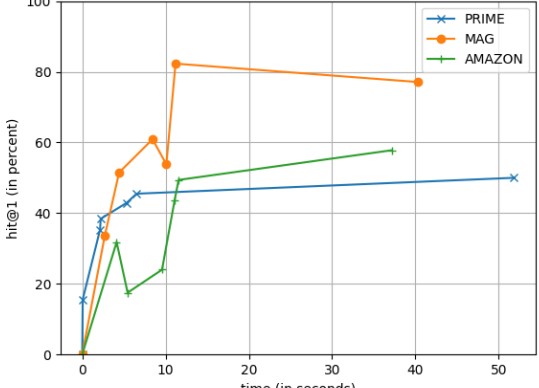 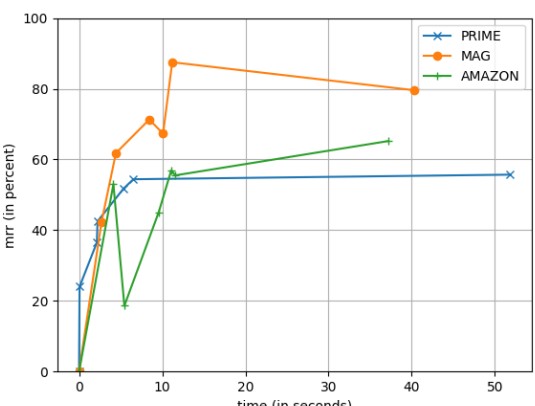

Figure 4: hit@1 (on the left) and mrr (on the right) for a) steps 1+7 b) steps 1-6 c) steps 1-7 d) steps 1-8 with pointwise reranking (parallel), e) steps 1-8 with listwise reranking, f) steps 1-8 with pairwise reranker, connected in this order.

Because of different strength (symbolic and embedding-based search), they effectively complement each other to an overall hit@20 rate of 71.9%.

**Latency**  Table 8 and Figure 4 give practical insights into the variants' latencies, measured on an EPYC 9454P (64 cores) CPU with subscription-free API access to the inference engine of Cerebras.ai. As expected, the regular expression-based step (step 3) is computationally negligible. Triplet grounding (step 5) terminates in less than 0.1 s in the majority of cases, but can take several seconds in complex cases with many triplets, reinforced with large $l_{max}$. A parallelization of loop iterations or a heuristic search of the lowest $l \leq l_{max}$ that returns $\geq k$ most likely answer candidates, could reduce the maximum runtime in some applications. The embedding-based steps complete fast on average with the PRIME SKB used, but take several seconds on the SKBs with millions of embedded entities of one node type. Overall, the runtime is dominated by LLM latency. The pointwise reranker is naturally parallelizable, and the median runtime is less than a second, while the pairwise reranker requires even several minutes in some cases. The implementation of a parallelizable sorting algorithm or heuristics has potential for speedup here. The listwise reranker strikes at balance and responds rapidly with suitable inference engines and a low reasoning effort setting.

## 5 Conclusion and Future Work

We introduced AF-Retriever, a modular and traceable framework for multi-hop question answering over Structured Knowledge Bases (SKBs). It interlinks several neural and symbolic components into a traceable workflow of established and novel approaches. These approaches include target type prediction, vector-semantic search with and without embedded relational knowledge, text-to-Cypher translation for standardized parsing, triplet-based graph exploration, and LLM-driven reranking. AF-Retriever is characterized by an incremental expansion strategy for the scope of constant-like entity candidates in its graph-based retrieval. This strategy balances sensitivity and specificity by searching for $k$ answers that satisfy all relational constraints while minimizing false positives. The system robustly handles both structured relations and unstructured textual information by sequentially coupling these components and incorporating a hybrid retrieval strategy.

Across diverse domains (biomedical, academic paper search, and product recommendation), AF-Retriever demonstrates strong empirical performance. It outperforms state-of-the-art results on the STaRK benchmarks regarding several metrics. Its design is modular, and, due to grounded triplets, AF-Retriever's answers are traceable. Modularity and traceability are two crucial features for deploying LLM-based QA systems in real-world scenarios. An extensive ablation study and evaluation of different compatible design choices highlight the contribution of each component. Notably, AF-Retriever can operate in a zero-shot setting, requiring neither task-specific training data nor fine-tuning, which makes it adaptable across domains.

Future work may explore the potential of fine-tuning individual components, such as Cypher generation and reranking, to achieve greater gains when training data for specific steps are available. Additionally, extending the framework to support expressive logical reasoning would increase its applicability. In summary, AF-Retriever and the modular evaluation of this work take an important step toward reliable and transparent high-precision question answering grounded in semi-structured knowledge.

### Acknowledgments

This research was partly funded by the German Federal Ministry of Research, Technology and Space under grant curAIknow (03ZU1202NA), part of project curATime (Cluster for Atherothrombosis and Individualized Medicine).

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

# A    Appendix

## A.1    STaRK SKB Properties

Table 9: Properties of the STaRK SKBs by Wu et al. (2024b)

|         | #entity types | #relation types | avg. degree | #entities | #relations | #tokens |
|---------|---------------|-----------------|-------------|-----------|------------|-------------|
| AMAZON  | 4             | 4               | 18.2        | 1,035,542 | 9,443,802  | 592,067,882 |
| MAG     | 4             | 4               | 43.5        | 1,872,968 | 39,802,116 | 212,602,571 |
| PRIME   | 10            | 18              | 125.2       | 129,375   | 8,100,498  | 31,844,769  |

## A.2  Used Prompts

**Prompt of DERIVE_TARGET_TYPE:**
Given several instances of these types: {candidate_types}. An instance of which type could correctly answer the query: {question}

Return nothing but the type of which the instance must be of.

**Prompt of DERIVE_CYPHER:**
Generate a Cypher query based on the given query Q. Please follow the restrictions precisely!

- Simple Syntax: Use a very basic and short Cypher syntax.

- Content Accuracy: Omit any information that cannot be exactly captured with one of the given, available node labels, or available keywords.

- No Quantifications: Avoid using quantifications.

- No Negations: Skip negated facts, avoid using "NOT" or "<>".

- No "OR": Do not use "OR".

- Available Keywords: Restrict yourself to the available keywords: MATCH, WHERE, RETURN, AND, CONTAINS.

- Date Format: Format dates as YYYY-MM-DD.

Given Information:

- Query Q: {query}

- Available Node Labels: {nodes_to_consider}

- Available Relationship Labels: {edges_to_consider}

Example: MATCH (d:disease)-[:is_effect/phenotype_of_disease]->(e:effect/phenotype)
MATCH (e)-[:protein/gene_is_associated_with_effect/phenotype]->(g:gene/protein)
WHERE g.name = "IGF1"
RETURN d.title

At the end of the query, RETURN y.title for the target (y:{target_type})
Only return one Cypher query, no additional information.

**Prompt of pointwise reranker:**
Examine if a {node_type} satisfies a given query and assign a score from 0.0 to 1.0. If the {node_type} does not satisfy the query, the score should be 0.0. If there exists explicit and strong evidence supporting that {node_type} satisfies the query, the score should be 1.0. If partial evidence or weak evidence exists, the score should be between 0.0 and 1.0. Here is the query: {query} Here is the information about the {node_type} Please score the {node_type} based on how well it satisfies the query. [. . . ]

**Prompt of listwise Reranker:**
The rows of the following list consist of an ID number, a type and a corresponding descriptive text: {possible_answers} Sort this list in descending order according to how well the elements can be considered as answers to the following query: {query} Please make absolutely sure that the element which satisfies the query best is the first element in your order. Return ONLY the corresponding ID numbers separated by commas in the asked order.'

**Prompt of pairwise Reranker:**

The following two elements consist of an ID number, a type and a corresponding descriptive text: {node1_id}, {node_type_1}, {doc_info_1}. {node2_id}, {node_type_2}, {doc_info_2}. Please find out which of the elements satisfy the following query better: {query} Return ONLY the corresponding ID number which corresponds to the element that satisfies the given query best. Nothing else.

### A.3 Triplet Grounding Algorithm

---

**Algorithm 2** GROUND_TRIPLETS

---

**Input:**

a set of triplets $\mathbb{T}$, a dictionary $\mathbb{S}$ that assigns a set of candidate nodes $\mathbb{S}(x)$ to each symbol $x$, a target variable $y$, a set of edges $\mathbb{E}$

**Output:** A set of candidate nodes for $y$

1: $\mathbb{S}_{clone} \leftarrow \mathbb{S}$
2: **for all** $\tau = (h, e, t) \in \mathbb{T}$ **do**
3:     $\mathbb{N} \leftarrow \emptyset$.
4:     **for all** $h^{(j)} \in \mathbb{S}(h)$ **do**
5:         $\mathbb{N} \leftarrow \mathbb{N} \cup \text{NEIGHBORS}(h^{(j)}, e, \mathbb{E}, -)$
6:     **end for**
7:     $\mathbb{S}(h) \leftarrow \mathbb{S}(h) \cap N$
8:     $\mathbb{N} \leftarrow \emptyset$.
9:     **for all** $t^{(j)} \in s(t)$ **do**
10:        $\mathbb{N} \leftarrow \mathbb{N} \cup \text{NEIGHBORS}(t^{(j)}, e, \mathbb{E}, +)$
11:     **end for**
12:     $\mathbb{S}(t) \leftarrow \mathbb{S}(t) \cap \mathbb{N}$
13: **end for**
14: **if** $\mathbb{S}_{clone} = \mathbb{S}$ **then**
15:     **return** $\mathbb{S}(y)$
16: **else**
17:     GROUND_TRIPLETS$(T, S, y, E)$
18: **end if**

---

## A.4 Base LLMs of Related Work

Table 10: Categorized benchmark methods developed for SKB-based multi-hop question answering and their LLMs employed. Section 4 contrasts their performance on the STaRK benchmark sets.

| retrieval category | method title | base LLM |
|---|---|---|
| textual | BM25: Best Matching 25 (Robertson et al., 1995)
VSS (Ada-002): Vector Similarity Search
VSS (Multi-ada-002): Multi-Vector Similarity Search
DPR: Dense Passage Retriever (DPR) (Karpukhin et al., 2020)
VSS + Reranker: Vector Similarity Search + LMM Reranker(Chia et al., 2024; Zhuang et al., 2024) | -
-
-
RoBERTa

GPT4 |
| structural | QA-Graph Neural Networks (Yasunaga et al., 2021)
ToG: Think-on-Graph (Sun et al., 2024) | -
- |
| hybrid | AvaTaR (Wu et al., 2024a)
4StepFocus (Boer et al., 2024)
KAR: Knowledge-Aware Retrieval (Xia et al., 2025)
HybGRAG: Hybrid Retrieval-Augmented Generation on Textual and Relational Knowledge Bases (Lee et al., 2025)
ReAct (Yao et al., 2023)
Reflexion (Shinn et al., 2023)
AF-Retriever (ours) | Claude 3 Sonnet
GPT4o
GPT4o
Claude 3 Opus

Claude 3 Opus
Claude 3 Opus
GPT OSS (120B) |
| hybrid with fine-tuned base LLM(s) | PASemiQA: Plan-Assisted Question Answering with Semi-structured data (Yang et al., 2025)
mFAR: Multi-Field Adaptive Retrieval (Li et al., 2024)
MoR: Mixture of Structural-and-Textual Retrieval (Lei et al., 2025)
GraphRAFT: Retrieval Augmented Fine-Tuning for Knowledge Graphs on Graph Databases | fine-tuned LLaMa2 (7B) + GPT4
custom
fine-tuned LLaMa 3.2 (3B)
fine-tuned gemma2-9b-text2cypher +OpenAI gpt-4o-mini +LLaMa-3.1 (8B) |

## A.5 Results on Human-generated Test Sets

Table 11: Performance of baseline, SOA methods, and AF-Retriever on the human-generated STaRK benchmark test sets. Methods for which we did not find a reporting for these datasets are omitted.

| dataset | PRIME | | | MAG | | | AMAZON | | |
|---|---|---|---|---|---|---|---|---|---|
| metric (in percent) | hit@1 | hit@5 | mrr | hit@1 | hit@5 | mrr | hit@1 | hit@5 | mrr |
| VSS (Ada-002) | 17.4 | 34.7 | 26.4 | 28.6 | 41.7 | 35.8 | 39.5 | 64.2 | 52.6 |
| VSS (Multi-ada-002) | 24.5 | 39.8 | 33.0 | 23.8 | 41.7 | 31.4 | 46.9 | _72.8_ | 58.7 |
| DPR | 2.0 | 9.2 | 7.0 | 4.7 | 9.5 | 7.9 | 16.0 | 39.5 | 27.2 |
| VSS + Reranker | 28.4 | 48.6 | 36.2 | 36.9 | 45.2 | 40.3 | 54.3 | _72.8_ | 62.7 |
| AvaTaR | 33.0 | 51.4 | 41.0 | 33.3 | 42.9 | 38.6 | 58.3 | **76.5** | _65.9_ |
| 4StepFocus | _50.5_ | _65.5_ | _57.9_ | 47.6 | 51.2 | 49.2 | _60.5_ | 67.9 | 64.3 |
| KAR | 45.0 | 60.6 | 51.9 | _51.2_ | _58.3_ | _54.5_ | **61.7** | _72.8_ | **66.3** |
| ReAct | 21.7 | 33.3 | 28.2 | 27.3 | 40.0 | 33.9 | 45.6 | 71.7 | 58.8 |
| Reflexion | 16.5 | 33.0 | 24.0 | 28.6 | 39.3 | 36.5 | 49.4 | 64.2 | 59.0 |
| AF-Retriever (ours) | **57.1** | **69.4** | **62.7** | **52.4** | **60.7** | **55.9** | 58.0 | 69.1 | 63.5 |

## A.6 Results of Extended Analysis

All experiments of this extended modular analysis were conducted on the same 10% of the QA pairs drawn from the validation sets. To avoid unnecessary computation, all metrics were measured before reranking in the following until Appendix A.6.7. Unless otherwise specified, the following parameters were used: $k = 20$, $\alpha = 2/3$, $l_{max} = 100$, LLM: GPT OSS 120B with "medium reasoning effort" for steps 1 and 2, and "low reasoning effort" for reranking, embeddings with relational information for step 7, and without relational information for steps 4 and 6.

### A.6.1 LLM choice for target type prediction and Cypher formalization

Table 12: Relative number of invalid and incorrect target type predictions after step 1 (direct instruction to return most likely target type) and step 2 (parsed from Cypher query) on PRIME.

| | step 1 | | step 2 | |
|---|---|---|---|---|
| | invalid answers | wrong target type | invalid answers | wrong target type |
| medium | **0.0** | **1.8** | **0.0** | 2.2 |

Table 13: Properties of different LLM's evaluated in Section 4.2 and in the rest of this subsection.

| model name | manufacturer | open weights | # parameters | context window |
|---|---|---|---|---|
| GPT-5 mini | OpenAI | proprietary | not disclosed | 400,000 |
| GPT OSS 120B | OpenAI | yes | 117B (5.1 activated) | 131.072 |
| LLaMa 4 Scout Instruct | Meta | yes | 109B (17B activated) | 327.680 |
| Gemma 3 27B | Google | yes | 27B | 131.072 |
| Mistral Small 3.2 Instruct | Mistral AI | yes | 24B | 128.000 |
| GPT OSS 20B | OpenAI | yes | 21B (3.6 activated) | 131.072 |
| Qwen 3 14.8B | Alibaba Cloud | yes | 15B | 40.960 |
| Phi-4 | Microsoft | yes | 14B | 16.384 |

Table 14: Relative number of invalid (unparseable syntax, e.g., more than one answer, or non-existent node type) and incorrect target type predictions after step 1 (direct instruction to return most likely target type) and step 2 (parsed from Cypher query) on PRIME for different open-weight LLMs. Most confused answers are "effect/phenotype" and "disease" as well as "pathway" and "biological process". MAG and AMAZON are omitted, because all answer candidates are of the same node type.

|  | step 1 | | step 2 | |
|---|---|---|---|---|
|  | invalid answers | wrong target type | invalid answers | wrong target type |
| GPT-5 mini | **0.0** | 3.1 | **0.0** | 1.8 |
| GPT OSS 120B | **0.0** | **1.8** | **0.0** | 2.2 |
| LLaMa 4 Scout Instruct | 3.6 | 6.3 | 0.4 | 4.0 |
| Gemma 3 27B | **0.0** | **1.8** | 0.9 | 6.2 |
| Mistral Small 3.2 Instruct | **0.0** | 3.1 | 1.3 | 7.6 |
| GPT OSS 20B | **0.0** | 2.2 | 4.9 | **1.3** |
| Qwen 3 14B | 0.9 | 3.1 | 2.2 | 4.0 |
| Phi-4 | **0.0** | 3.6 | 2.7 | 6.7 |

Table 15: Relative number of invalid and incorrect target type predictions after step 1 (direct instruction to return most likely target type) and step 2 (parsed from Cypher query) on PRIME for different "reasoning effort" values. Evaluated on GPT OSS 120B LLM.

|  | step 1 | | step 2 | |
|---|---|---|---|---|
|  | invalid answers | wrong target type | invalid answers | wrong target type |
| none | **0.0** | 2.7 | **0.0** | 1.8 |
| low | **0.0** | **1.8** | **0.0** | **0.9** |
| medium | **0.0** | **1.8** | **0.0** | 2.2 |
| high | **0.0** | 2.2 | 4.9 | 1.3 |

Table 16: Performance of AF-Retriever with different LLMs.

| | PRIME | | MAG | | AMAZON | |
|---|---|---|---|---|---|---|
| metrics (in percent) | hit@20 | recall@20 | hit@20 | recall@20 | hit@20 | recall@20 |
| GPT-5 mini | 70.5 | 49.7 | 95.1 | 57.9 | **77.3** | **32.6** |
| GPT OSS 120B | **71.9** | **52.2** | **95.9** | **58.4** | 75.3 | 31.9 |
| LLaMa 4 Scout Instruct | 58.0 | 41.6 | 85.7 | 51.2 | 72.1 | 30.5 |
| Gemma 3 27B | 65.2 | 45.1 | 89.5 | 54.2 | 75.3 | 31.2 |
| Mistral Small 3.2 Instruct | 59.8 | 42.4 | 93.6 | 57.2 | 74.7 | 31.3 |
| GPT OSS 20B | 69.2 | 49.3 | 94.7 | 57.6 | **77.3** | 32.2 |
| Qwen 3 14B | 59.8 | 41.7 | 92.1 | 56.3 | 75.3 | 31.5 |
| Phi-4 | 67.4 | 48.8 | 95.1 | 57.6 | 73.4 | 30.5 |

Table 17: Performance of AF-Retriever with different "reasoning effort" hyperparameters of the LLM (GPT OSS 120B).

| | PRIME | | MAG | | AMAZON | |
|---|---|---|---|---|---|---|
| metrics (in percent) | hit@20 | recall@20 | hit@20 | recall@20 | hit@20 | recall@20 |
| none | 68.3 | 49.3 | **95.9** | 58.3 | 76.0 | 32.0 |
| low | 68.3 | 49.2 | 95.1 | 57.8 | 76.6 | 32.0 |
| medium | **71.9** | **52.2** | **95.9** | **58.4** | 75.3 | 31.9 |
| high | 70.5 | 50.1 | 95.5 | 58.0 | **77.3** | **32.6** |

### A.6.2   Strict vs. lenient parsing mode

Table 18: Effect of lenient parsing mode (no hard filter by node and edge types) vs strict parsing mode (ruling out all triplets with invalid node or edge types) and their hybrids. Concerns step 3.

|  | PRIME | | MAG | | AMAZON | |
|---|---|---|---|---|---|---|
|  | hit@20 | recall@20 | hit@20 | recall@20 | hit@20 | recall@20 |
| NO filter by node and edge types | 69.2 | 50.6 | 86.1 | 52.0 | 74.0 | 31.5 |
| Filter triplets by relation types | 70.5 | 51.7 | **95.9** | **58.4** | **76.0** | **32.1** |
| Filter constants by node types | **71.9** | **52.2** | **95.9** | **58.4** | 75.3 | 31.9 |
| Filter by both labels | 70.1 | 50.9 | **95.9** | **58.4** | 75.3 | 31.9 |

### A.6.3 Constant candidate scope expansion

Table 19: This table summarizes how AF-Retriever performs under varying $l_{\max}$ values. If steps 1–3 yield fewer than one relevant triplet linked to the target entity, steps 4–6 are skipped, as triplet-based retrieval cannot be performed. The first column reports the proportion of QA pairs processed in steps 4 and 5. The next two columns present the average precision (fraction of true positives among all answer candidates) and average specificity (fraction of true positives among all ground-truth answers) for those QA pairs. Smaller $l_{\max}$ values typically result in fewer grounded answers. The following column indicates the number of QA pairs for which at least one answer candidate was retrieved. This number provides an upper bound for hit@m for step 6 ($\forall m$) shown in the next column. The final two columns display the overall performance of AF-Retriever for different $l_{\max}$ values, including supplemented answers from step 7. All metrics represent average relative scores, expressed as percentages. The three vertical sections refer to the PRIME, MAG, and AMAZON validation sets.

| $l_{max}$ | steps 1-5 | | | steps 1-6 | | | steps 1-8 | |
|---|---|---|---|---|---|---|---|---|
| | processed | precision | specificity | >0 retrievals | hit@20 | recall@20 | hit@20 | recall@20 |
| 0 | 0.0 | - | - | - | - | - | 46.4 | 33.0 |
| 1 | 87.9 | **24.8** | 48.6 | 55.4 | **71.8** | **51.8** | 69.2 | 49.4 |
| 2 | 87.9 | 23.0 | 54.1 | 64.3 | 67.4 | 50.7 | 71.0 | 51.2 |
| 3 | 87.9 | 21.6 | 56.1 | 67.4 | 68.2 | 50.6 | 71.0 | 51.2 |
| 10 | 87.9 | 18.4 | 59.0 | 73.7 | 67.3 | 49.5 | 71.0 | 51.6 |
| 33 | 87.9 | 12.9 | 61.4 | 79.0 | 64.4 | 46.9 | 71.4 | 51.8 |
| 100 | 87.9 | 7.0 | **64.3** | 82.6 | 64.9 | 47.4 | **71.9** | **52.2** |
| 333 | 87.9 | 6.5 | **64.3** | **83.0** | 64.5 | 47.2 | **71.9** | **52.2** |
| 0 | 0.0 | - | - | - | - | - | 72.6 | 44.9 |
| 1 | 85.0 | **30.4** | 92.9 | 83.5 | 93.7 | 53.5 | 95.9 | 58.2 |
| 2 | 85.0 | **15.5** | 94.2 | 83.5 | 95.5 | 53.9 | 95.5 | 58.0 |
| 3 | 85.0 | 11.4 | 89.9 | 82.4 | 92.0 | 55.3 | 94.9 | **60.8** |
| 10 | 85.0 | 9.0 | 96.0 | 84.2 | 96.0 | 55.1 | **96.2** | 58.6 |
| 33 | 85.0 | 8.8 | 96.4 | 84.6 | 96.4 | 55.2 | 95.9 | 58.4 |
| 100 | 85.0 | 8.8 | **96.8** | **85.0** | **96.5** | **55.4** | 95.9 | 58.4 |
| 333 | 85.0 | 8.8 | **96.8** | **85.0** | **96.5** | **55.4** | 95.9 | 58.4 |
| 0 | 0.0 | - | - | - | - | - | 74.7 | 30.5 |
| 1 | 64.9 | **5.7** | 35.3 | 42.9 | **60.6** | **26.7** | **77.9** | **32.6** |
| 2 | 64.9 | 4.1 | 36.6 | 47.4 | 53.4 | 24.9 | 76.0 | 32.1 |
| 3 | 64.9 | 4.9 | 39.0 | 52.6 | 55.6 | 24.3 | 77.3 | 32.3 |
| 10 | 64.9 | 3.8 | 42.0 | 55.2 | 55.3 | 23.6 | 77.3 | 32.3 |
| 33 | 64.9 | 3.9 | 43.5 | 60.4 | 50.5 | 22.5 | 75.3 | 31.9 |
| 100 | 64.9 | 3.8 | **44.3** | **62.3** | 50.0 | 22.1 | 75.3 | 31.9 |
| 333 | 64.9 | 3.8 | **44.3** | **62.3** | 50.0 | 22.1 | 75.3 | 31.9 |

### A.6.4 Embeddings

Table 20: Differential analysis of adding or leaving out relational information in textual node descriptions before embedding each SKB node for VSS for step 4 (all symbol candidates), step 6 (graph-based retrieval), and step 7 (VSS-based retrieval). The relational information includes direct neighbors and 2-hop paths where the relation type to the embedded entity is 1-1 or n-1.

|  | relations included in | | | | | | | |
|  | step 4 | step 6 | step 7 | hit@1 | hit@5 | h@20 | r@20 | mrr |
|---|---|---|---|---|---|---|---|---|
| PRIME | yes | yes | yes | 28.1 | 48.7 | 67.4 | 49.0 | 36.9 |
|  | yes | yes | no | 26.3 | 46.0 | 67.0 | 48.8 | 35.5 |
|  | yes | no | yes | 29.0 | 49.6 | 68.3 | 49.5 | 38.1 |
|  | yes | no | no | 27.2 | 46.9 | 67.4 | 48.8 | 36.6 |
|  | no | yes | yes | _37.9_ | _55.4_ | _71.0_ | _51.7_ | 44.9 |
|  | no | yes | no | 37.1 | 53.6 | 70.1 | 51.3 | 44.1 |
|  | no | no | yes | **38.4** | **56.2** | **71.9** | **52.2** | **45.8** |
|  | no | no | no | 37.5 | 54.5 | 70.5 | 51.3 | _45.0_ |
| MAG | yes | yes | yes | 61.3 | 86.5 | 95.5 | 58.2 | 72.4 |
|  | yes | yes | no | **62.0** | 86.5 | 94.7 | 57.9 | 73.0 |
|  | yes | no | yes | 60.9 | 86.5 | 94.7 | 57.8 | 72.5 |
|  | yes | no | no | 61.7 | 86.5 | 94.0 | 57.6 | 73.1 |
|  | no | yes | yes | 61.3 | **87.2** | **96.6** | **58.7** | 72.7 |
|  | no | yes | no | **62.0** | **87.2** | _95.9_ | _58.4_ | _73.3_ |
|  | no | no | yes | 60.9 | **87.2** | _95.9_ | 58.4 | 72.8 |
|  | no | no | no | 61.7 | **87.2** | 95.1 | 58.1 | **73.4** |
| AMAZON | yes | yes | yes | **24.7** | _42.9_ | 76.6 | _33.2_ | **33.6** |
|  | yes | yes | no | _24.0_ | 41.6 | **77.9** | **33.7** | _33.4_ |
|  | yes | no | yes | _24.0_ | **43.5** | 76.0 | 32.6 | 33.1 |
|  | yes | no | no | 23.4 | 42.2 | _77.3_ | 33.0 | 32.9 |
|  | no | yes | yes | 23.4 | 42.2 | 76.0 | 32.6 | 32.7 |
|  | no | yes | no | 22.7 | 40.9 | _77.3_ | 33.0 | 32.5 |
|  | no | no | yes | _24.0_ | _42.9_ | 75.3 | 31.9 | 32.8 |
|  | no | no | no | 23.4 | 41.6 | 76.6 | 32.4 | 32.5 |

### A.6.5 Omitting ambiguous node types

Table 21: Effect of concealing specific node types and incident edge types in the LLM prompt for receiving the Cypher query (step 2).

|  | concealed types | h@20 | r@20 |
|---|---|---|---|
| MAG | ∅ | _88.0_ | _52.2_ |
|  | "field of study" | **95.9** | **58.4** |
| AMAZON | ∅ | _75.3_ | _31.9_ |
|  | "category" | **77.9** | **33.4** |

### A.6.6   Weighting of both parallel retrieval strategies

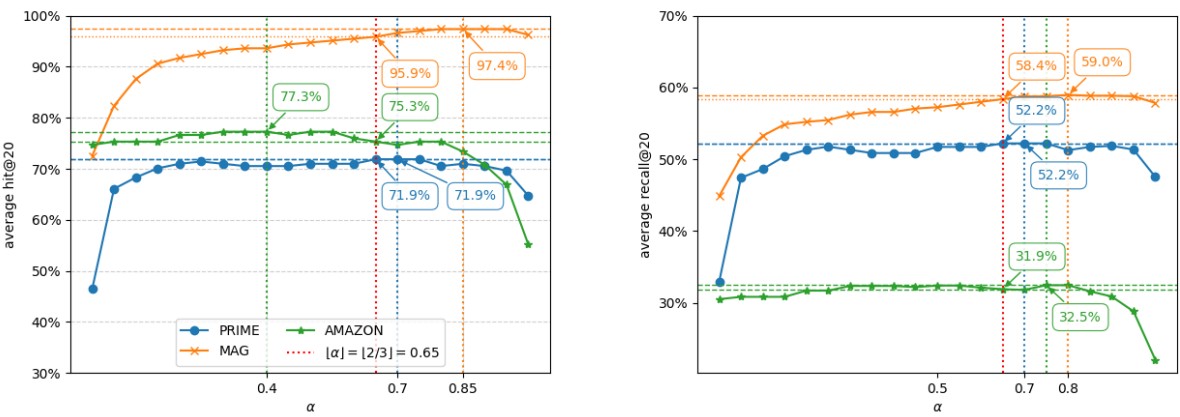

Figure 5: Hit@20 and recall@20 of AF-Retriever depending on $\alpha$ on 10% of the validation sets

### A.6.7   Results reranker

Table 22: Performance of AF-Retriever with different reranking strategies.

| reranker | PRIME | | | MAG | | | AMAZON | | |
|---|---|---|---|---|---|---|---|---|---|
| | hit@1 | hit@5 | mrr | hit@1 | hit@5 | mrr | hit@1 | hit@5 | mrr |
| pointwise | 41.1 | 64.3 | 51.7 | 55.3 | 81.2 | 67.3 | 48.1 | 68.2 | 56.9 |
| listwise | 47.8 | 62.5 | 54.4 | **83.1** | **93.6** | **87.5** | 47.4 | 64.9 | 55.5 |
| pairwise | **49.1** | **64.7** | **55.7** | 71.8 | 89.5 | 79.6 | **60.4** | **72.1** | **65.2** |

## A.7 Error propagation

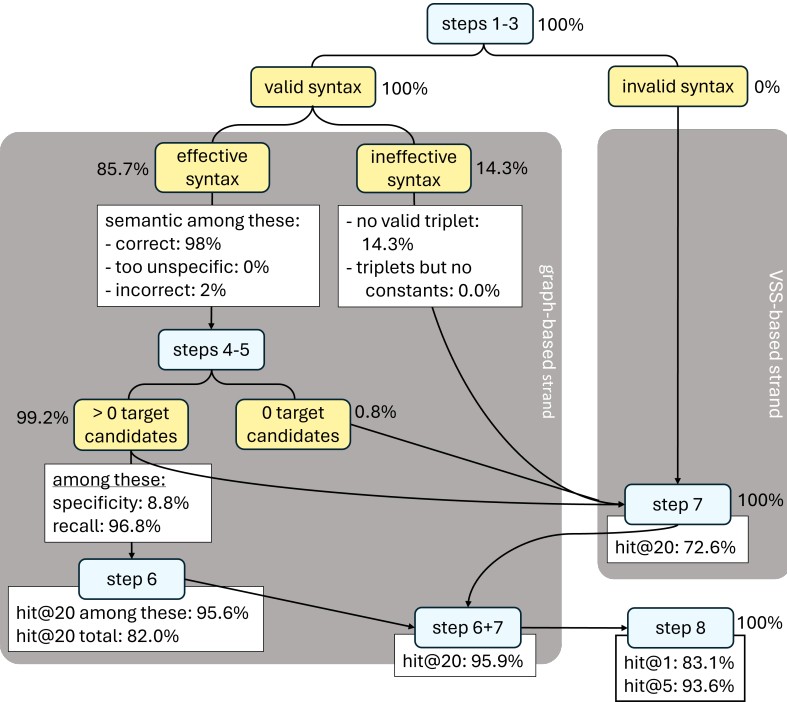

Figure 6: Error propagation in MAG

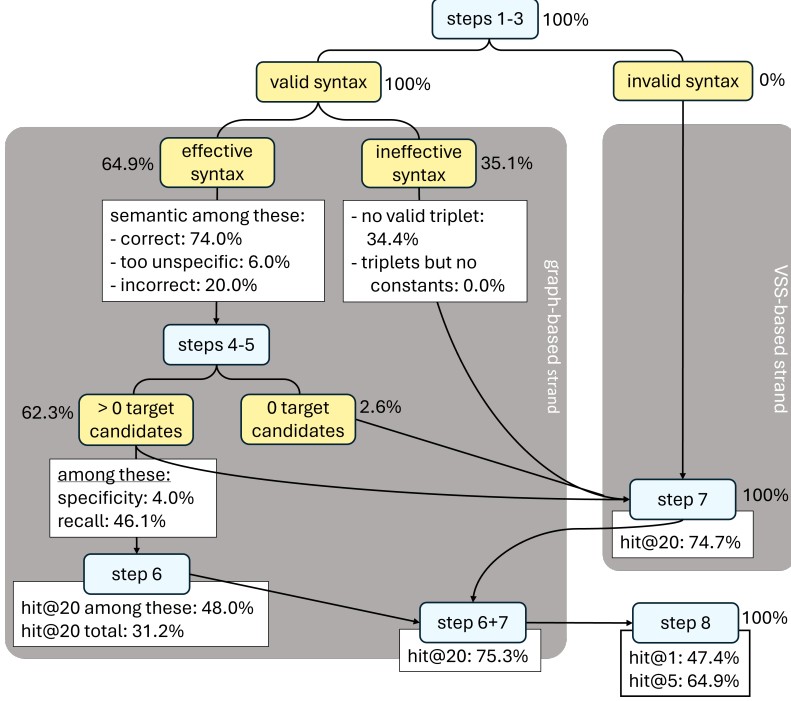

Figure 7: Error propagation in AMAZON

Table 23: Stereotypical examples of PRIME test questions and predicted Cypher queries (step 2) for each manually assigned semantic quality category in the error propagation analysis

| semantic rating | question | generated query |
|---|---|---|
| correct semantic | Could you recommend a drug that targets the DPYD gene/protein and synergizes with Rosiglitazone to improve treatment outcomes? Also, I'm looking for options with protein binding around 52% for dosage considerations. | MATCH (g:gene/protein name: "DPYD"), (r:drug name: "Rosiglitazone"), (y:drug)-[:drug_targets_ gene/protein]->(g), (y)-[:drug_synergistically_ interacts_with_drug]->(r) WHERE y.protein_binding = 52 RETURN y.title |
| » comment: Specific, correct query. | | |
| too unspecific | What is the name of the pediatric condition associated with the APOA1 gene that causes severe reduction of high-density lipoproteins in blood and is marked by enlarged liver and spleen, swollen lymph nodes, oversized tonsils, nerve issues, and possible heart complications in adults? incorrect semantic | MATCH (g:gene/protein)-[:protein/gene_is_ associated_with_disease]->(d:disease) WHERE g.name = "APOA1" RETURN d.title |
| » comment: All phenotypes/symptoms are missing. | | |
| incorrect semantic | Is lithium involved in modulating the activity of genes or proteins that bind to phosphatidylinositol-4,5-bisphosphate, consequently inhibiting calcium buildup in cardiac cells and thereby diminishing the likelihood of arrhythmias? | MATCH (y:drug)-[:drug_targets_gene/protein]->(g:gene/protein) WHERE y.name = "Lithium" RETURN y.title |
| » comment: Lithium is not a gene/protein and further relations are ignored. | | |

Table 24: Stereotypical examples of MAG test questions and predicted Cypher queries (step 2) for each manually assigned semantic quality category in the error propagation analysis

| semantic rating | question | generated query |
|---|---|---|
| correct semantic | Find publications by coauthors of "High-pressure phases of Weyl semimetals NbP, NbAs, TaP and TaAs" on the topic of topological and superconducting characteristics. | MATCH (p:paper title: "High-pressure phases of Weyl semimetals NbP, NbAs, TaP and TaAs")<-[:author_writes_paper]-(a:author)-[:author_writes_paper]->(y:paper) WHERE y.title CONTAINS "topological" AND y.title CONTAINS "superconducting" RETURN y.title |
| » comment: Specific and correct query. | | |
| too unspecific | n/a | n/a |
| incorrect semantic | Looking for research papers from Karlsruhe Institute of Technology on the topic of numerical stress distribution models in a 2017 high-field accelerator magnet within the context of Conductor studies. | MATCH (i:institution name: "Karlsruhe Institute of Technology")<-[:author_affiliated_with_institution]-(a:author)-[:author_writes_paper]->(y:paper) WHERE y.year = 2017 AND y.title CONTAINS "numerical stress distribution" AND y.title CONTAINS "high-field accelerator magnet"AND y.title CONTAINS "Conductor" RETURN y.title |
| » comment: 2017 refers to the magnet, not the publication data. | | |

Table 25: Stereotypical examples of AMAZON test questions and predicted Cypher queries (step 2) for each manually assigned semantic quality category in the error propagation analysis

| semantic rating | question | generated query |
|---|---|---|
| correct semantic | What are some fun and uniquely designed swim goggles that would pair well with my Bestway Swimming Goggles High Style 3 Per Order? | MATCH (x:product)-[:product_also_buy_product]->(y:product) WHERE x.title = "Bestway Swimming Goggles High Style 3 Per Order" AND y.title CONTAINS "fun" AND y.title CONTAINS "unique" RETURN y.title |
| » comment: Specific and correct. | | |
| too unspecific | What are some home exercise machines that can complement the Total Gym 1600 in terms of range of exercises? Also, what are the popular workout machines that fitness enthusiasts usually pair with the Total Gym Weight Bar? | MATCH (s:product)-[:product_also_buy_product]->(y:product) WHERE s.name CONTAINS "Total Gym" RETURN y.title |
| » comment: Too vague. The searched product should contain "Total Gym Weight Bar" | | |
| incorrect semantic | Are there any compact running belts with good construction that can serve as a suitable replacement for the Sporteer Kinetic K1? | MATCH (x:product)-[:product_also_buy_product]->(y:product) WHERE x.title = "Sporteer Kinetic K1" AND y.category = "running belt" AND y.size = "compact" AND y.construction = "good" RETURN y.title |
| » comment: The relation type should be product_also_view_product for products to replace. product_also_buy_product rather fits for accessories. | | |

