# OpenReview forum: "Autofocus Retrieval: An Effective Pipeline for Multi-Hop Question Answering With Semi-Structured Knowledge"
_TMLR — Accepted by TMLR_

### Review · Reviewer_vpkS · 2026-01-20

**Summary Of Contributions:**

The paper proposes **Autofocus-Retriever (AF-Retriever)**, a modular framework designed for multi-hop question answering over Semi-Structured Knowledge Bases (SKBs), which contain both structured knowledge graphs and unstructured text. The authors introduce a novel "incremental scope expansion" mechanism that dynamically adjusts the number of candidate entities to satisfy relational constraints, mimicking an optical autofocus process.

The pipeline consists of four main phases: (1) Target type prediction and Cypher query generation via LLMs; (2) An "autofocus" retrieval step that grounds triplets and expands search scope; (3) A hybrid retrieval strategy combining graph-based and vector similarity search (VSS); and (4) A final LLM-based reranking step (comparing pointwise, pairwise, and listwise strategies).

**Strengths:**

*
**Strong Empirical Performance:** The method achieves state-of-the-art results on the STaRK benchmarks (PRIME, MAG, AMAZON) in zero-shot and one-shot settings, significantly outperforming existing baselines like VSS, KAR, and even some fine-tuned models like GraphRAFT on specific metrics.


*
**Rigorous Ablation Study:** The authors provide a comprehensive analysis of individual components, demonstrating the value of the hybrid retrieval strategy and the specific contributions of the reranking module.


*
**Practicality:** The approach does not require domain-specific fine-tuning, making it highly adaptable for real-world cold-start scenarios.



**Weaknesses:**

*
**High Latency:** As acknowledged by the authors, the pipeline (especially the reranking step) incurs significant latency, which may limit its applicability in real-time use cases.


*
**Complexity:** The pipeline involves multiple distinct steps and LLM calls, increasing the potential for error propagation (e.g., if the initial Cypher generation fails).

**Audience:**

Yes

**Audience Explanation:**

The problem of reasoning over **semi-structured data** (hybrid text and graph) is of high interest to the TMLR community, particularly for researchers working on Retrieval-Augmented Generation (RAG) and Knowledge Graphs.

1. **Relevance:** As RAG systems move beyond simple vector similarity, methods that can leverage structured constraints without requiring expensive model training are highly valuable.
2. **Methodological Interest:** The "incremental scope expansion" algorithm  offers a novel perspective on balancing precision and recall in retrieval, which could inspire further research in neuro-symbolic integration.

**Broader Impact Concerns:**

No concerns. The authors address the potential for reducing hallucinations via grounding. The work does not introduce new ethical concerns beyond those inherent to the use of Large Language Models.

**Claims And Evidence:**

Yes

**Claims Explanation:**

The authors provide convincing evidence to support their claims. They evaluate the proposed AF-Retriever on three diverse datasets from the STaRK benchmark, covering biomedical, academic, and e-commerce domains.

1.
**Comparison with Baselines:** The paper compares the method against a wide range of baselines, including traditional methods (BM25, VSS), agent-based methods (ReAct), and recent SKB-specific approaches (KAR, GraphRAFT). The improvement in Hit@1 (e.g., surpassing the second-best method by ~32% on average) is substantial and clearly presented.


2.
**Ablation Studies:** The breakdown of performance in Table 4  clearly isolates the gains from the graph-based retrieval, the VSS supplement, and the reranking step, justifying the architectural choices.


3.
**Detailed Analysis:** The authors do not shy away from discussing limitations, such as the runtime analysis in Table 19, which adds to the credibility of the evaluation.

**Requested Changes:**

I lean towards accepting this paper as it presents a robust, effective solution to a relevant problem. However, I have a few suggestions that would strengthen the submission:

1. **Latency vs. Accuracy Trade-off (Strengthening):**
The paper mentions that the full pipeline, especially with pairwise/listwise reranking, is computationally expensive (Table 19). It would be beneficial to include a brief discussion or a plot showing the trade-off between latency and accuracy. For instance, how much performance is lost if a faster (pointwise) reranker is used, or if the "scope expansion" limit () is reduced? This would help practitioners decide how to deploy the model in latency-sensitive environments.


2. **Error Propagation Analysis (Strengthening):**
The reliance on Step 2 (Cypher query generation)  is critical. While Table 9  analyzes target type prediction errors, could the authors provide more insight into cases where the LLM generates syntactically valid but semantically incorrect Cypher queries? A qualitative analysis of "failure cases" where the graph retrieval path fails entirely (forcing reliance on the VSS fallback) would be insightful.


3. **Clarification on "Scope Expansion" (Minor):**
In Step 5, the incremental increase of  (candidates per constant) is described. Please clarify if this process is parallelized or sequential in the implementation, as this significantly impacts the "worst-case" runtime for complex queries.

---

> ### Author Response · Authors · 2026-03-02
> **Answer to Reviewer vpkS**
>
> Dear Reviewer vpkS,
>
> Thank you for your very fast but careful assessment. We addressed your concerns as follows.
>
> (1) Latency vs. Accuracy Trade-off (Strengthening): We added a dedicated paragraph in Subsection 4.2 analyzing the latency-accuracy trade-off. The revised version includes a new plot (Figure 4) that directly contrasts performance gains with end-to-end latency across variants. To improve accessibility, we moved the latency table from the Appendix to Subsection 4.2 (now Table 8), positioning it next to the reranking results for direct comparison. This makes the cost-quality trade-off explicit and easier to interpret.
>
> (2) Error Propagation Analysis (Strengthening): We incorporated another new paragraph in Subsection 4.2 that quantifies error propagation across pipeline stages. A new figure (Figure 3 in the main text for one representative dataset, remaining datasets in the Appendix A.7) visualizes the error propagation path and labels it with statistics at each stage. In addition, we performed a qualitative semantic analysis by manually categorizing generated Cypher queries and providing representative failure cases in the Appendix A.7. This clarifies where errors originate and how they affect downstream components.
>
> (3) Clarification on "Scope Expansion" (Minor): Step 5 (incremental candidate expansion per constant) is currently sequential and not parallelized. At the end of Section 4.2, paragraph “Latency”, we added a short note discussing potential optimization strategies to reduce the maximum runtime.

---

> > ### Comment · Reviewer_vpkS · 2026-03-31
> > **Acknowledgment of Revisions and Final Recommendation**
> >
> > Dear Authors,
> > Thank you for your detailed and constructive response to my review. You have thoroughly addressed my concerns.Latency vs. Accuracy: Moving the latency breakdown (Table 8) to the main text and adding the trade-off plots (Figure 4) perfectly clarifies the deployment practicalities and performance trade-offs.Error Propagation: The new flowcharts (Figure 3, Figures 6 and 7) and the qualitative semantic analysis (Tables 23-25) clearly map out the pipeline's failure modes and add valuable interpretability.Scope Expansion: Thank you for clarifying that Step 5 is currently sequential and acknowledging the potential for future parallelization to optimize runtime.The paper now presents a highly robust and transparent framework. I am happy to maintain my recommendation to Accept the paper.

---

### Review · Reviewer_s2zz · 2026-02-06

**Summary Of Contributions:**

The paper proposes a retrieval pipeline for multi-hop question answering. It uses LLMs to predict answer types and translate questions into Cypher-style constraints, retrieves candidates with vector similarity, grounds relational triplets using an incremental scope expansion procedure, mixes graph-constrained candidates with additional unconstrained dense retrieval to improve robustness, and then applies LLM reranking to select the final top results. It reports strong zero-shot or one-shot performance on STaRK QA benchmarks and includes ablations indicating that hybrid retrieval and reranking are key contributors to the gains.

Strengths:
1. The paper proposes a clear, modular pipeline that integrates symbolic constraints with dense retrieval, and it motivates why hybridization reduces susceptibility to single-step errors.
2. The empirical results are strong and include ablations that connect improvements to specific pipeline components.
3. The reranking discussion is concrete and compares multiple LLM reranking paradigms rather than assuming a single default.

Weaknesses:
1. Several claimed advantages hinge on LLM behavior in intermediate steps (type prediction and Cypher generation). The paper notes sensitivity in certain domains, but the main text would benefit from clearer reporting on robustness and failure modes in these steps, beyond the PRIME discussion.
2. Comparisons across methods can be difficult to interpret because baselines differ in supervision and fine-tuning, and the paper itself highlights that some stronger results come from fine-tuned systems on specific datasets.

**Audience:**

Yes

**Audience Explanation:**

The work is relevant to readers interested in retrieval over hybrid graph-plus-text corpora, multi-hop QA.

**Claims And Evidence:**

Yes

**Claims Explanation:**

The paper supports performance claims with benchmark comparisons and provides an ablation study that demonstrates incremental gains from hybrid retrieval and reranking.

**Requested Changes:**

1. Clarify evaluation comparability across baselines. The main results table mixes zero-shot or one-shot systems and fine-tuned systems, and the narrative already notes that fine-tuning can change outcomes substantially. The paper should more explicitly standardize the comparison conditions or clearly separate the conclusions that pertain only to zero-shot or one-shot methods.

2. Strengthen reporting on intermediate-step reliability. Since the pipeline depends on LLM-generated Cypher and extracted constraints, add concise quantitative statistics in the main text about Cypher validity, parsing success rates, and how often downstream retrieval fails due to upstream extraction errors, ideally broken down by dataset.

3. Improve the description of hyperparameter selection and sensitivity. The method uses parameters such as $\alpha$, and the text mentions that optimizing $\alpha$ on validation could improve results. Please move a compact sensitivity summary into the main paper, including how default values were chosen and how sensitive performance is across datasets.

4. Add a concise cost and latency accounting. Since the approach makes multiple LLM calls plus reranking, provide a main-text estimate of average number of LLM calls per query, typical token usage for each reranking strategy, and a rough runtime profile so readers can assess practicality.

---

> ### Author Response · Authors · 2026-03-02
> **Response to Reviewer s2zz**
>
> Thank you for your detailed suggestions. We have implemented the following revisions.
>
> (1) Clarify evaluation comparability across baselines: We restructured the results section (Section 4.1) and split the main results table to clearly separate zero-shot/one-shot methods from fine-tuned systems. The revised text more clearly separates the conclusions that pertain only to zero-shot and one-shot methods now, without, disregarding fine-tuned systems (despite the limitation of transferability).
>
> (2) Strengthen reporting on intermediate-step reliability: We incorporated a new paragraph in Subsection 4.2 that quantifies error propagation across pipeline stages in addition to the ablation study. A new figure (Figure 3 in main text for one representative dataset, remaining datasets in the Appendix A.7) visualizes the error propagation path and labels it with statistics at each stage. In addition, we performed a qualitative semantic analysis by manually categorizing generated Cypher queries and providing representative failure cases in Appendix A.7. This clarifies where errors originate and how they affect downstream components.
>
> (3) Improve the description of hyperparameter selection and sensitivity: We expanded the “Further Analyses” section (Section 4.2) and structurally separated main results, ablation study, hyperparameter sensitivity, error propagation, and latency analysis. The hyperparameter subsection now includes a labeled sensitivity plot (Figure 2) showing retrieval performance as a function of $\alpha$, along with a comparison of the default and optimal settings. The description of the experimental settings of the main results section (Section 4.1) is supplemented with short explanations for the choices of $l_{max}$, the LLM, the reranking strategy, and a forward reference to the sensitivity of $\alpha$.
>
> (4) Cost and latency accounting: We added a dedicated paragraph in Subsection 4.2 detailing per-component latency, total runtime, and LLM usage. The latency table (now Table 8) and reranking statistics (number of LLM calls per query and typical token consumption, now Table 7) were moved from the Appendix to the main text. A new plot (Figure 4) visualizes the latency-accuracy trade-off to make cost implications explicit.

---

### Review · Reviewer_34vi · 2026-02-17

**Summary Of Contributions:**

The paper targets multi-hop QA over semi-structured knowledge bases  that link structured graphs (nodes/edges) with unstructured documents. It demonstrate strong performance on the STaRK QA benchmarks. The main contributions are (1) incremental scope expansion to improve grounding, (2) an weighted hybrid retrieval that mixes Cypher-based candidates with pure vector search, and (3) a systematic comparison of LLM reranking strategies.

**Audience:**

Yes

**Audience Explanation:**

The paper presents a practical retrieval pipeline for multi-hop QA over semi-structured knowledge, with some interesting observations in the experimental results (especially the main results part). These findings should be useful to researchers working on retrieval, RAG, and knowledge-grounded QA.

**Broader Impact Concerns:**

I don’t see major ethical or broader-impact concerns specific to this work.

**Claims And Evidence:**

Yes

**Claims Explanation:**

(1) The paper is well written and easy to follow. Figures, tables, and examples in each step help understand the main designs.

(2) The paper provides strong experimental evidence on multiple datasets and includes a careful step-by-step ablation study showing the impact of each component, which makes the main claims convincing.

(3) The appendix includes detailed analyses on multiple aspects of the method, including prompt design, comparisons across different LLM backbones before reranking for the AF-Retriever, and a thorough study of reranking strategies and latency.

**Requested Changes:**

(1) The approach relies heavily on model choices. Replicability across different LLMs is discussed in the appendix but without reranking. Could you provide a small table showing performance with other widely available LLMs for the whole pipeline?

(2) Novelty is not clear. Many building blocks (text-to-Cypher prompting, VSS, symbolic grounding, LLM reranking) are known, and the method’s uniqueness rests mainly on incremental scope expansion and the particular hybridization. Please clearly state which parts are novel algorithmic contributions and engineering integration.

(3) It would be helpful to include a cost–quality plot comparing the three reranking strategies, given their very different token and latency profiles.

---

> ### Author Response · Authors · 2026-03-02
> **Response to Reviewer 34vi**
>
> Dear Reviewer 34vi,
>
> Thank you for your constructive feedback. We have revised the manuscript accordingly.
>
> (1) Performance with other widely available LLMs: In Subsection 4.2 (paragraph hyperparameter sensitivity), we added a table (Table 6) reporting full-pipeline performance with different LLMs. This provides a clearer picture of robustness across alternatives.
>
> (2) Novelty clarification: We inserted an additional paragraph between the discussion of SKBs and the description of AF-Retriever in Section 1. This paragraph explicitly delineates the conceptual and methodological differences from prior work and clarifies the original contributions of our framework.
>
> (3) Cost-quality trade-off: As suggested, we added a dedicated latency and cost analysis in Subsection 4.2, including a plot (Figure 4) that visualizes the trade-off between latency and accuracy across variants. We also moved the latency table and reranking statistics (including LLM calls and token usage) into the main text (now Table 6) to support direct interpretation.

---

### Decision · Action_Editor_bWao · 2026-04-02

**Recommendation:** Accept as is

**Additional Comments:**

Reviewer 34vi suggested improving Figure 2 for readability and presentation.

**Audience:**

Yes

**Audience Explanation:**

The proposed retrieval pipeline for multi-hop QA over semi-structured knowledge is practical and can be useful for people working on areas involving retrieval, RAG, multi-hop QA, knowledge-grounded QA.

**Claims And Evidence:**

Yes

**Claims Explanation:**

Reviewers commonly agree that this paper is well written and the performance is strong. The claims are well supported by the detailed ablation studies, error propagation analysis, and detailed discussions. The approach is also practically adaptable without requiring domain-specific fine-tuning. All the reviewers acknowledged that their concerns have been addressed.